# Functional genomics reveal gene regulatory mechanisms underlying schizophrenia risk

Yongxia Huo[1], Shiwu Li[1,2], Jiewei Liu[1], Xiaoyan Li[1,2] & Xiong-Jian Luo [1,2,3,4]

Genome-wide association studies (GWASs) have identified over 180 independent schizophrenia risk loci. Nevertheless, how the risk variants in the reported loci confer schizophrenia susceptibility remains largely unknown. Here we systematically investigate the gene regulatory mechanisms underpinning schizophrenia risk through integrating data from functional genomics (including 30 ChIP-Seq experiments) and position weight matrix (PWM). We identify 132 risk single nucleotide polymorphisms (SNPs) that disrupt transcription factor binding and we find that 97 of the 132 TF binding-disrupting SNPs are associated with gene expression in human brain tissues. We validate the regulatory effect of some TF binding-disrupting SNPs with reporter gene assays (9 SNPs) and allele-specific expression analysis (10 SNPs). Our study reveals gene regulatory mechanisms affected by schizophrenia risk SNPs (including widespread disruption of POLR2A and CTCF binding) and identifies target genes for mechanistic studies and drug development. Our results can be accessed and visualized at SZDB database (http://www.szdb.org/).

[1] Key Laboratory of Animal Models and Human Disease Mechanisms of the Chinese Academy of Sciences & Yunnan Province, Kunming Institute of Zoology, Chinese Academy of Sciences, Kunming, Yunnan 650223, China. [2] Kunming College of Life Science, University of Chinese Academy of Sciences, Kunming, Yunnan 650204, China. [3] Center for Excellence in Animal Evolution and Genetics, Chinese Academy of Sciences, Kunming 650223, China. [4] KIZ-CUHK Joint Laboratory of Bioresources and Molecular Research in Common Diseases, Kunming Institute of Zoology, Chinese Academy of Sciences, Kunming Yunnan 650223, China. These authors contributed equally: Yongxia Huo, Shiwu Li. Correspondence and requests for materials should be addressed to X.-J.L. (email: luoxiongjian@mail.kiz.ac.cn)

Schizophrenia (SCZ) is a severe psychiatric disorder characterized by positive symptoms, negative symptoms and cognitive impairments. With high lifetime prevalence (~0.5–1%), substantial morbidity and mortality, SCZ poses a major threat to global health. The pathogenesis of SCZ remains elusive. However, accumulating data suggest that inherited genetic variants play critical roles in SCZ (the heritability of SCZ is estimated around 0.8[1]). To date, genome-wide association studies (GWASs) have identified over 180 loci that show strong association with SCZ[2–14]. Nevertheless, for most of the risk loci, the causal variant(s) and the mechanisms by which the risk variants exert their effects on SCZ remain unknown.

Most of the SCZ risk variants identified by GWAS are located in non-coding regions[12], implying that these variants exert their effects through altering gene expression. Consistently, recent studies have shown that schizophrenia-associated variants are significantly enriched in regulatory regions[15,16], suggesting that disruption of regulatory function may represent a common mechanism that non-coding genetic variants confer risk of SCZ. Though accumulating evidence support the hypothesis that most of the risk variants identified by GWAS contribute to SCZ risk through affecting gene expression rather than protein structure or function, only very limited functional variants have been identified so far[17–19]. Due to the complexity of linkage disequilibrium (LD) and gene regulatory, identifying the functional (or causal) variants (at each reported locus) and elucidating their regulatory mechanisms remain major challenges in psychiatric genetics.

Regulatory elements (REs) (e.g., promoters and enhancers) are non-coding DNA sequences that have a critical role in controlling gene expression. Previous studies have shown that most of associations identified by GWASs were attributable to variants located in REs[12,20]. In fact, investigating if the risk variants are localized in REs has been proved a useful way to identify causal (or functional) variants for complex diseases[21,22]. Regulatory elements usually contain multiple binding sites (i.e., DNA motifs that can be recognized by transcription factors (TFs)) for TFs and genetic variations in REs affect gene expression through altering binding affinity of TFs. The architecture and function of REs are variable in different tissues and previous studies have shown that the activity of REs have strong tissue and cellular specificity[23,24]. Thus, understanding the tissue-specific structure and activity of REs is crucial to elucidate how genetic variants contribute to risk of SCZ through affecting the regulatory function of REs.

Here we perform a systematic and deep analysis to identify the functional variants and to elucidate the gene regulatory mechanisms underlying the genetic associations reported by recent schizophrenia GWASs[12–14]. We identify functional variants at multiple risk loci through integrating a wide range of data from high-throughput functional genomics experiments, including genome-wide binding landscapes (chromatin immunoprecipitation and sequencing (ChIP-Seq)) for 30 TFs and position weight matrices (PWMs). We identify 132 functional single-nucleotide polymorphisms (SNPs) that disrupt TF binding and investigate the gene regulatory mechanisms of these TF binding–disrupting SNPs. We validate the regulatory effect of several identified functional SNPs using reporter gene assays and allele-specific expression analysis. In addition, we identify the potential target genes of the identified regulatory SNPs by using brain expression quantitative trait loci (eQTL) from three independent eQTL datasets. Finally, we show that nervous system development-related genes are significantly enriched among the target genes of the TF binding–disrupting SNPs, providing further support for the neurodevelopmental hypothesis of schizophrenia. The results reported in this study can be visualized and downloaded at SZDB (http://www.szdb.org/)[25].

## Results

### Prioritization of functional SNPs and experimental validation.

Potential causal risk SNPs from three large-scale GWASs[12–14] were used in this study. The first GWAS was from the schizophrenia working group of the psychiatric genomics consortium (PGC2)[12]. PGC performed a large-scale GWAS of SCZ (36,989 cases and 113,075 controls) and reported 108 independent risk loci[12]. The second GWAS was from the study of Li et al.[13] which identified 30 new SCZ risk loci recently through combining the association results from the Chinese and PGC2 samples. The third GWAS was from a recent study of Pardinas et al.[14] that reported 50 novel risk loci for SCZ. As each risk locus contains hundreds of SNPs that showed similar association significance (due to LD), potential causal SNPs were identified by PGC2 and Pardinas et al.[14]. PGC2 identified set of SNPs (i.e., potential causal set of SNPs) that were 99% likely to contain the causal variants[12]. Pardinas et al.[14] also identified potential causal SNPs using FINEMAP[26]. A total of 18,707 potential causal SNPs from PGC2 and 1799 potential causal SNPs (spanning 50 novel risk) from the study of Pardinas et al.[14] were used in this study. For the 30 new risk loci reported by Li et al.[13], we performed LD analysis and identified 4794 SNPs that were in LD with the index SNPs ($r^2 > 0.3$). The potential causal SNPs (from PGC2[12] and Pardinas et al.[14]) and SNPs in LD with the index SNPs (from Li et al.[13]) are likely to contain the causal variants. In total, 23,400 non-overlapping potential causal SNPs from above three GWASs were included in this study (Fig. 1). To pinpoint the potential causal variant (or variants) at each locus, we systematically annotated the credible causal SNPs through combining bioinformatic and high-throughput functional genomics analyses (Fig. 1). We first annotated the potential causal SNPs with well-characterized functional annotation approaches (CADD[27], GWAVA[28], Eigen[29], RegulomeDB[30] and LINSIGHT[31]) and identified the most possible functional SNPs at each risk locus (Supplementary Data 1 and Supplementary Tables 1 and 2). Of note, two different strategies were used by the functional annotation methods to prioritize the potential functional SNP. For CADD, Eigen, GWAVA and LINSIGHT, the larger the score, the higher probability that the SNP is functional. Therefore, the SNP with the highest score was defined as the top (i.e., the most likely) functional SNP. For RegulomeDB, smaller rating suggests higher probability that the SNP is functional. Thus, the SNP with the smallest rating was defined as the top functional SNP. For CADD, Eigen, GWAVA and LINSIGHT, the SNP with the highest score at each locus was defined as the top functional SNP. For RegulomeDB, the SNP with the smallest rating was defined as the top functional SNP. For each locus, we compared the top functional SNPs identified by different annotation methods and found that 153 loci (PGC2[12] performed regular LD clumping (implemented in PLINK[32]) to define the index SNPs (with following parameters: $r^2 < 0.1$, $P1 < 5 \times 10^{-8}$, window size <500 kb) and the genomic region containing all SNPs that were in LD (i.e., $r^2 > 0.6$) with each of the 128 index SNPs was defined as a locus) contained overlapping top functional SNPs prioritized by at least two different annotation methods (i.e., at least two methods annotated the same SNP as the most possible functional or causal SNP) (Supplementary Data 1 and Supplementary Tables 1 and 2), suggesting these SNPs were promising functional SNPs. Most of the prioritized functional SNPs were located in intergenic and intronic regions (Fig. 2a).

To identify the potential target genes regulated by the prioritized top functional SNPs (i.e., top SNPs identified by at least two different annotation approaches), we examined the associations between the prioritized top functional SNPs and gene expression in three brain eQTL databases (including the Lieber Institute for Brain Development (LIBD) eQTL browser[33]

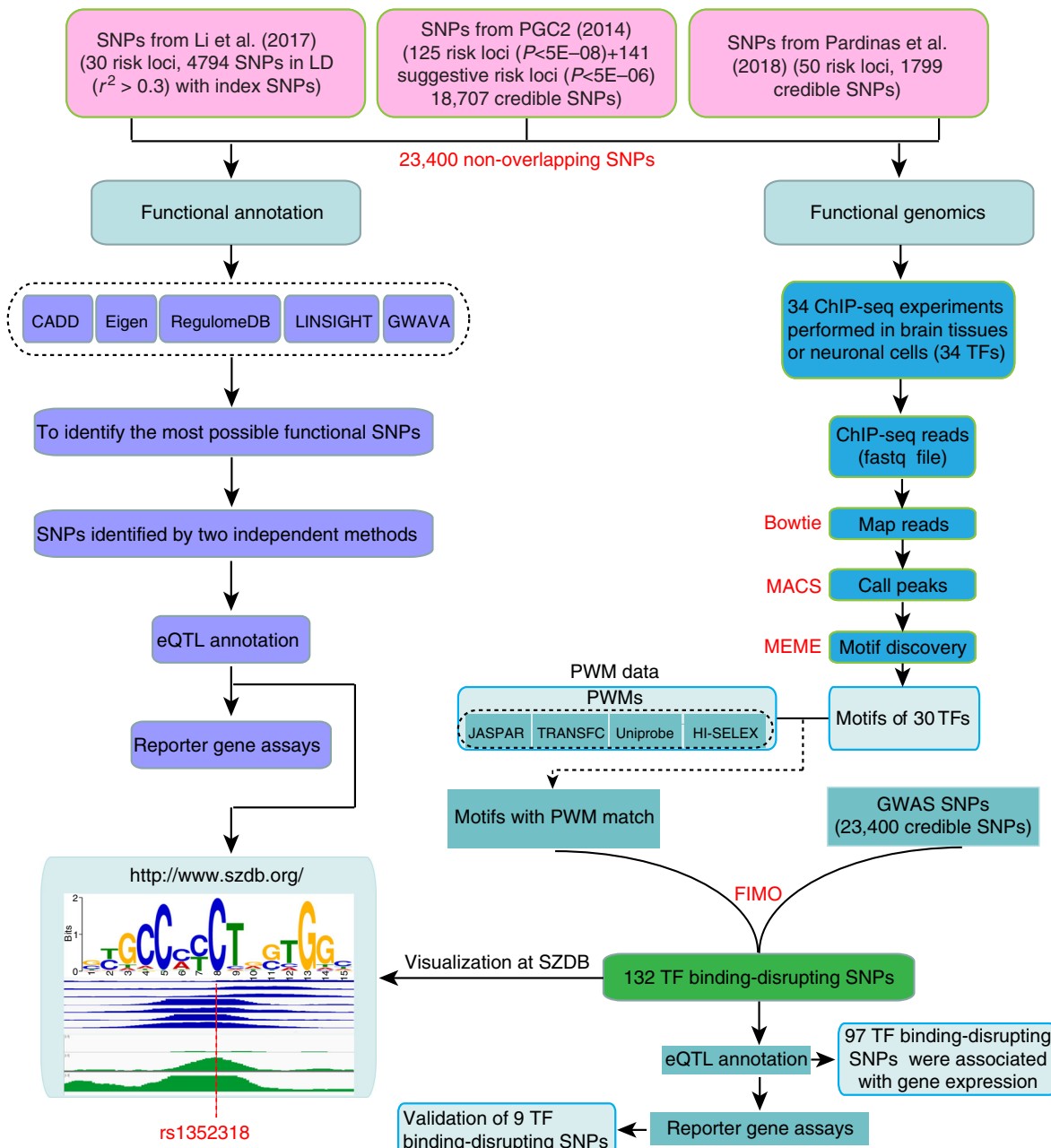

**Fig. 1** Prioritization and identification of regulatory single-nucleotide polymorphisms (SNPs) at schizophrenia risk loci. Risk SNPs (including credible causal SNPs and SNPs that were in linkage disequilibrium (LD) with the index risk SNPs) from three large-scale genome-wide association studies (GWASs) were subjected to functional annotation and functional genomics analyses. Five annotation methods were used to prioritize the most possible functional (or causal) SNPs (i.e., SNPs with the largest scores (CADD, Eigen, LINSIGHT and GWAVA) or the lowest rankings (1a, RegulomeDB)). The most possible functional SNPs (we call these SNPs top functional SNPs) were further distilled and the same top functional SNP prioritized by at least two different annotation approaches was subjected to expression quantitative loci (eQTL) annotation and reporter gene assays. We also utilized functional genomics to identify the potential causal SNPs at the risk loci. Chromatin immunoprecipitation and sequencing (ChIP-Seq) experiments performed in brain tissues or neuronal cell lines were used to identify the motifs (i.e., position weight matrices (PWMs) of corresponding transcription factors (TFs)). The identified PWMs from ChIP-Seq experiments and PWM database were compared, and the matched PWMs were used to map if the risk SNPs were located in the identified PWMs. Reporter gene assays were used to validate the effects of the identified TF binding–disrupting SNPs and eQTL annotation was performed to identify the potential target genes of the identified regulatory SNPs. The results of this study can be accessed and visualized at SZDB database (http://www.szdb.org/)

(dorsolateral prefrontal cortex, $N = 412$, including 175 patients with schizophrenia and 237 unaffected controls) (http://eqtl.brainseq.org/phase1/eqtl/), the Genotype-Tissue Expression (GTEx)[34] (tissues were from 13 brain regions, $N$ ranges from 80 to 154) (Supplementary Table 3) and the CommonMind Consortium (CMC)[35] (dorsolateral prefrontal cortex, $N = 467$).

In total, 66 prioritized top functional SNPs showed significant association (default false discovery rates (FDRs) used in the original papers were used in this study, i.e., FDR < 0.05 in CMC, FDR < 0.01 in LIBD). We used $P < 0.001$ for GTEx) with gene expression in human brain tissues (Supplementary Data 2). We validated the regulatory effects of 10 prioritized top functional

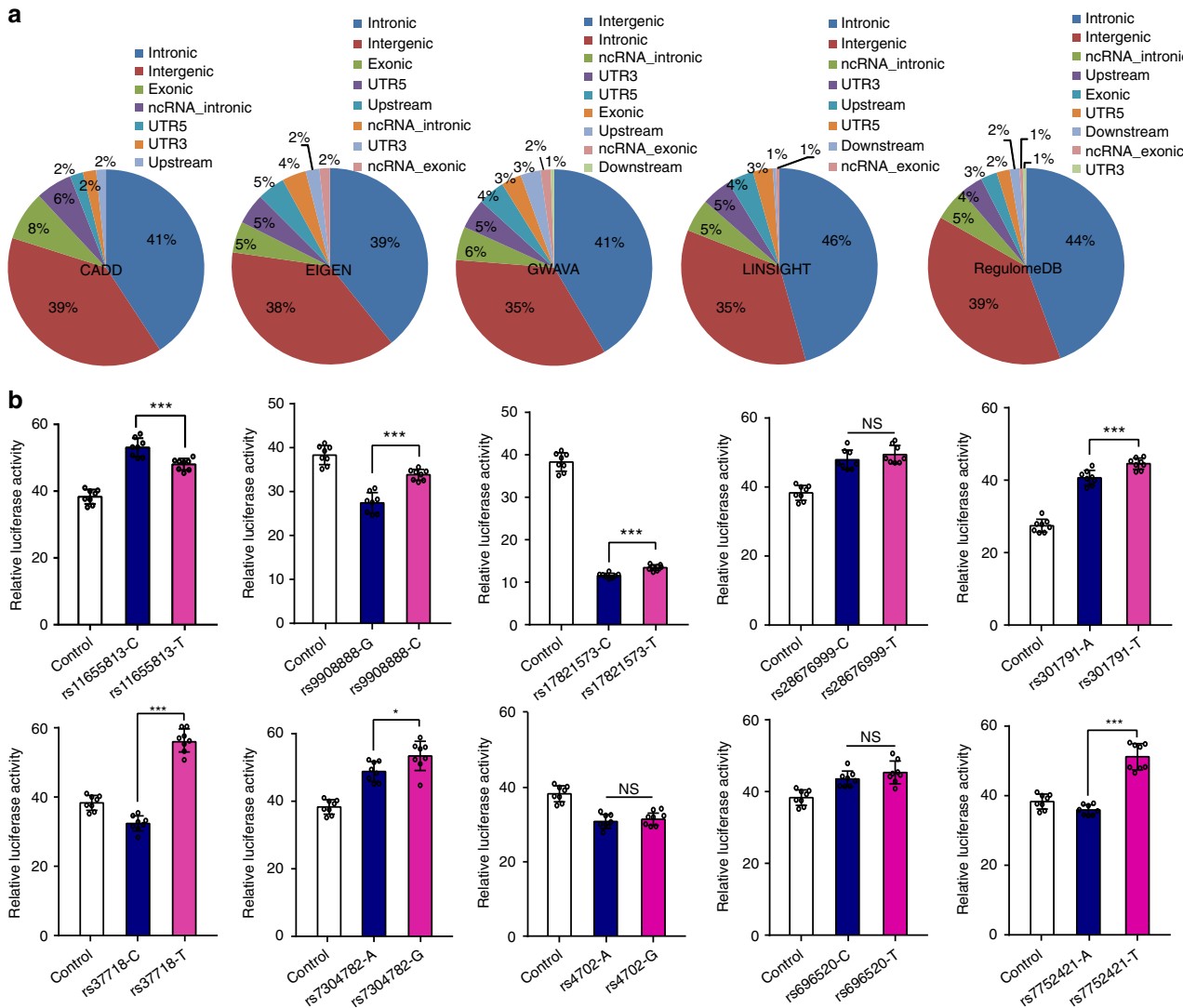

**Fig. 2** The distribution of the prioritized top functional single-nucleotide polymorphisms (SNPs) in genome and validation of the regulatory effects of the prioritized top functional SNPs. **a** Most of the prioritized top functional SNPs were located in intronic and intergenic regions. **b** Results of the reporter gene assays. Among the 10 tested SNPs, different alleles at 7 SNPs affected the expression of reporter gene significantly in HEK293 cells ($P < 0.05$). Data are expressed as mean ± SD from 8 technical replicates. NS, not significant, *$P < 0.05$, ***$P < 0.001$. Two-tailed Student's $t$-test. Source data are provided as a Source Data file

SNPs using reporter gene assays (Methods). The selection criteria of top functional SNPs for reporter gene assays were as follows. First, this SNP was prioritized as the top functional SNP (has the highest score or the smallest rating) at least by two different annotation methods simultaneously. Second, this SNP was associated with gene expression in human brains (Supplementary Data 2). We tested 10 SNPs and found that 7 out of 10 prioritized top functional SNPs (including rs11655813, rs9908888, rs17821573, rs301791, rs37718, rs7304782 and rs7752421) showed significant allelic effects in reporter gene assays in HEK 293 cells (i.e., different alleles at these SNPs affected the expression of reporter gene significantly) ($P < 0.05$, two-tailed Student's $t$-test, not corrected for multiple testing) (Fig. 2b). For these 7 significant SNPs, we compared the allelic effects on luciferase activity observed in reporter gene assays with the estimated eQTL effects of the corresponding alleles in CMC dataset. Among these 7 significant SNPs, 5 SNPs (rs11655813, rs9908888, rs17821573, rs301791 and rs7752421) were significantly associated (FDR < 0.01) with gene expression in CMC dataset. For these 5 significant eQTL SNPs, we found that the

allelic effects on luciferase activity observed in reporter gene assays are consistent with the estimated eQTL effects of the corresponding alleles in CMC dataset (Supplementary Fig. 1). These results prioritized the most possible functional SNPs at each risk locus and suggested that these functional SNPs may confer risk of schizophrenia through modulating gene expression.

**Identification of regulatory risk SNPs using functional genomics**. Though promising functional SNPs have been prioritized by the well-characterized annotation approaches (Supplementary Data 1 and Supplementary Tables 1 and 2), there are several limitations for these annotation methods. First, sequence conservation is an important factor for some annotation approaches (including CADD, Eigen and LINSIGHT). However, considering that some of the schizophrenia-associated variants are located in human-specific (or accelerated) region[36,37] (e.g., a recent study showed that a human-specific tandem repeat in intronic region of *CACNA1C* have pivotal role in schizophrenia susceptibility[36]), it is challenging for these annotation methods to prioritize the

potential functional SNPs located in these human-specific regions. Second, despite genomic features such as ChIP-Seq, TF binding sites and eQTL were used by some of these annotation methods, the tissues and cell types used to derive these genomic features were usually not from the brain or neuronal-related tissues. Considering that many of the regulatory variants have strong tissue-specificity, the potential functional SNPs prioritized by these annotation approaches may not be functional in human brains. Third, even the potential functional SNP prioritized by these annotation methods have functional consequences in brain or neuronal-related tissues, and the exact gene regulatory mechanisms (e.g., binding of which transcription factor was disrupted by the potential functional SNPs) of these potential functional SNPs remain unknown.

To further investigate how schizophrenia risk variants affect gene expression and elucidate the regulatory mechanisms of the potential causal variants, we annotated the potential causal SNPs by combining ChIP-Seq and PWM data (Fig. 1), as described previously[38]. We focused on risk variants that have regulatory effect in human brain tissues (or neuronal cells) as schizophrenia is thought to be a disorder that is mainly originated from dysfunction of brain function. First, most schizophrenia risk genes (including genes involved in neurotransmission (e.g., *DRD2*, *GRM3*, *GRIN2A*, *CACNA1C* and *CACNA1I*)) identified by genetic studies play pivotal roles in brain[12,14]. Second, accumulating evidence supports the neurodevelopmental hypothesis of schizophrenia[39,40]. Consistently, studies have showed that schizophrenia risk genes (including *DISC1*, *RELN* and *GLT8D1*) have important role in brain development through regulating proliferation and differentiation of neural stem cells[41–43]. More important, a recent study showed that schizophrenia associations were strongly enriched at enhancers active in brain tissues[12]. These lines of evidence suggest that schizophrenia risk variants exert their effects mainly in brain tissues. Therefore, brain tissues (or neuronal cells) may represent the most relevant tissues for studying the effects of SCZ risk variants. We thus used ChIP-Seq and eQTL data from brain tissues (or neuronal cells) to identify the potential causal SNPs and target genes regulated by the identified TF binding–disrupting SNPs. We downloaded and processed 34 ChIP-Seq experiments (Supplementary Table 4) from ENCODE[44]. After quality control, ChIP-Seq data of 30 TFs were retained and the genome-wide binding sites of each TF were identified using the ChIP-Seq data. We derived the DNA binding motifs of individual TF using the genomic binding locations from the ChIP-Seq data (Methods)[45]. DNA binding motifs derived from ChIP-Seq experiments were then compared with PWM data (from PWM databases) (Methods) and the matched PWM was used for further analysis (as EZH2 PWM results derived from ChIP-Seq were inconsistent with the PWM databases, we chose the most significant PWM (i.e., with the smallest *E*-value) from ChIP-Seq for further analysis). Potential causal SNPs were mapped to the matched PWMs and risk SNPs residing in PWMs were identified. To test if a risk SNP (located in a given PWM) disrupts TF binding, we used FIMO (Find Individual Motif Occurences)[46] to analyze the genomic sequences containing two different alleles of the given SNP (Methods). By doing so, we could identify TFs whose binding are disrupted by credible causal risk SNPs at single-base resolution by combining ChIP-Seq and corresponding PWM data.

Through annotating 23,400 potential causal risk SNPs with ChIP-Seq and PWM data, we identified 132 SNPs that disrupted the binding of 21 distinct TFs (Fig. 3a and Supplementary Data 3). Each of these SNPs was located in a TF binding motif that overlapped with a PWM match (*P* < 0.001, *P* was converted from log-odds scores, assuming a zero-order background model) on one or both alleles (http://www.szdb.org/)[25]. These TF

binding–disrupting SNPs (which we call regulatory SNPs) are located in DNA binding motifs that have strong nearby ChIP-Seq signals, indicating that corresponding TFs can bind to the genomic regions containing these SNPs in human brain tissues or neuronal cell lines. Over 70% TF binding–disrupting SNPs were located in intronic and intergenic regions (Fig. 3a). To compare the distribution of TF binding–disrupting SNPs identified in this study and random SNPs, we sampled random SNPs and annotated their genomic locations using ANNOVAR[47] (see Methods). We found that 56% random SNPs were located in intergenic regions (Supplementary Fig. 2). However, only 21% TF binding–disrupting SNPs were located in intergenic regions. By contrast, 50% TF binding–disrupting SNPs and only 35% random SNPs were located in intronic regions, suggesting that causal risk variants for schizophrenia were enriched in intronic regions. Nevertheless, more work is needed to verify this observation. Among the 132 TF binding–disrupting SNPs, 40 were located in POLR2A (RNA polymerase II subunit A) binding motifs and 38 were located in CTCF binding sites, suggesting widespread disruption of POLR2A and CTCF binding by SCZ risk SNPs (Fig. 3a). We noticed that some regulatory SNPs disrupted binding of two or more TFs simultaneously (Fig. 3b). The binding motif (PWM), ChIP-Seq signal, DNase-Seq signal and histone modification marks for each of the 132 SNPs can be accessed and visualized at SZDB (http://www.szdb.org/).

**Disruption of POLR2A binding by regulatory SNPs.** Forty SCZ risk SNPs disrupted binding of POLR2A (Fig. 3a), implying disruption of POLR2A binding may represent a common mechanism that schizophrenia risk variants exert their effect. We investigated how a risk SNP (rs1801311) at 22q13.2 disrupted POLR2A binding. Genetic variants at 22q13.2 showed significant association with SCZ in recent GWAS[12] (Supplementary Fig. 3). However, due to the complexity of LD, it is challenging to pinpoint the causal (or functional) SNP at this locus. Through integrating ChIP-Seq and PWM data, we identified a functional SNP (rs1801311) at 22q13.2. rs1801311 was located in a well-characterized binding motif for POLR2A (Fig. 4a). Strong POLR2A ChIP-Seq signal clearly showed the binding of POLR2A to the genomic region containing rs1801311 in human brain tissues and neuronal cell lines. Consistently, DNase-Seq and histone modification data indicated that rs1801311 was located in an actively regulatory region (Fig. 4b). We further validated the regulatory effect of rs1801311 using reporter gene assays (Fig. 4c–e). Genomic sequences (638 bp) containing different alleles of rs1801311 were cloned into pGL3-promoter vector to modulate the expression of luciferase reporter gene (Methods). The A allele of rs1801311 conferred a significant higher activity compared with G allele in all three tested cell lines (HEK293, SK-N-SH and SH-SY5Y) (*P* < 0.001, two-tailed Student's *t*-test), supporting the regulatory effect of rs1801311. Collectively, the data from ChIP-Seq, PWM, DNase-Seq, histone modification and reporter gene assays consistently support that rs1801311 may exert its regulatory effect through disrupting POLR2A binding.

**Disruption of CTCF binding by regulatory SNPs.** In addition to POLR2A, binding of CTCF and other TFs were also frequently disrupted by SCZ risk SNPs (Fig. 3a). CTCF binding was disrupted by 38 SCZ risk SNPs, including rs12912934 (15q25.2), rs16937 (1q32.1) and rs7012106 (8q24.3) (Figs. 5 and 6a). These three regulatory SNPs (i.e., rs12912934, rs16937 and rs7012106) were located in CTCF binding sites and allelic differences at these SNPs disrupted the binding of CTCF. DNase-Seq data showed that these three SNPs were located in transcriptionally active

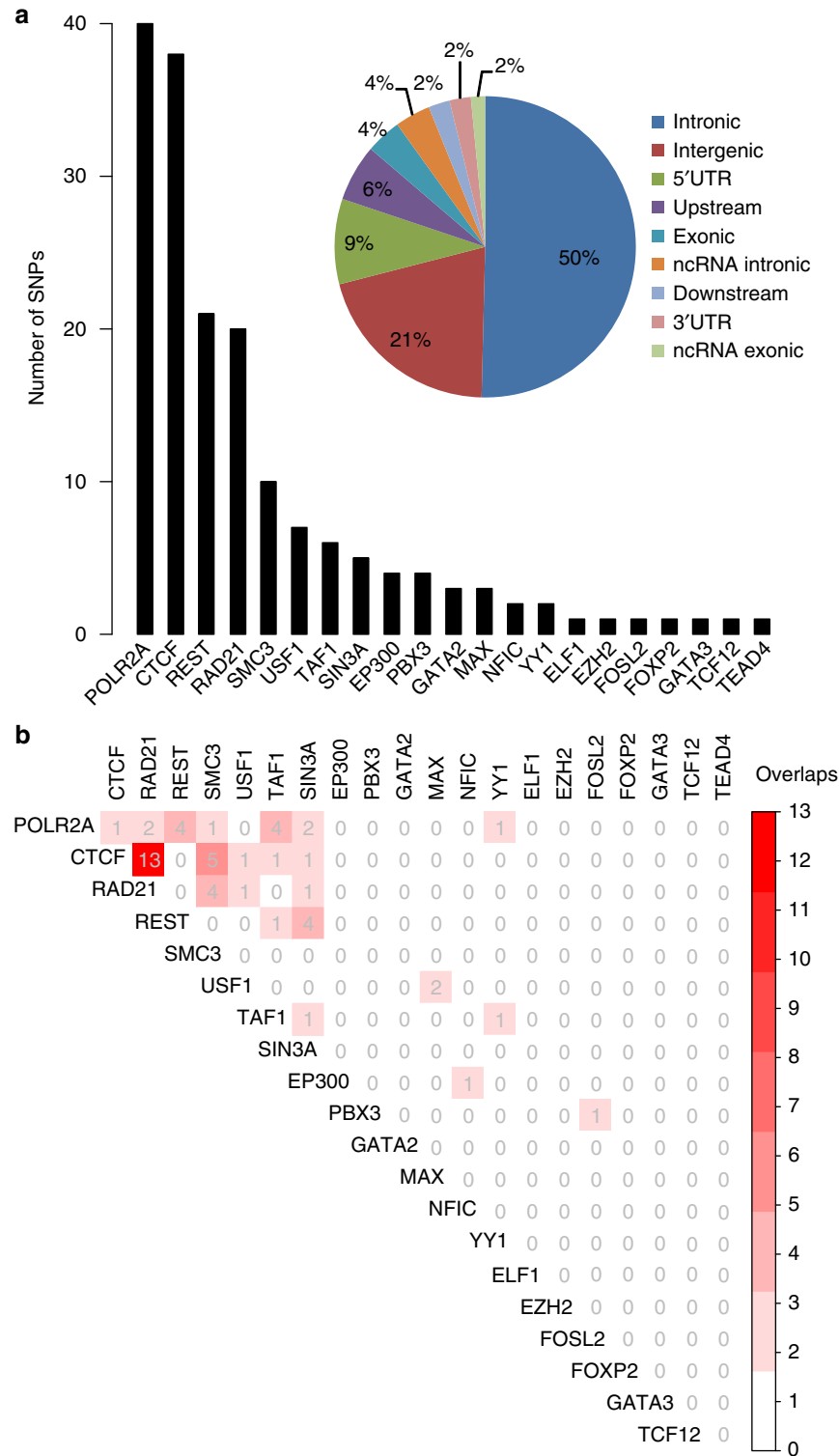

**Fig. 3** Transcription factors (TFs) disrupted by schizophrenia risk single-nucleotide polymorphisms (SNPs). **a** Left panel, The box plot shows the number of schizophrenia risk SNPs disrupting the binding of individual TF based on matched position weight matrices (PWMs) from chromatin immunoprecipitation and sequencing (ChIP-Seq) and PWM database. **a** Right panel, The distribution of the 132 identified TF binding–disrupting SNPs in the human genome. Most of the identified regulatory SNPs were located in intronic and intergenic regions. **b** The number of SNPs disrupting the binding of two TFs simultaneously. In addition to disruption of individual TF, we also found that some of the identified regulatory SNPs disrupt the binding of two TFs simultaneously. Of note, 13 risk SNPs disrupted the binding of CTCF and RAD21 simultaneously

regions. ChIP-Seq data indicated that CTCF could bind to the genomic regions containing these three regulatory SNPs in human brain tissues or neuronal cell lines (Fig. 5). We further verified the regulatory effect of these three SNPs with reporter gene assays. We found that the T allele of rs12912934 conferred a significant higher reporter gene activity compared with C allele in SK-N-SH and SH-SY5Y cells ($P < 0.05$, two-tailed Student's $t$-test) (Fig. 5c, d). For SNP rs16937, the G allele conferred a significant

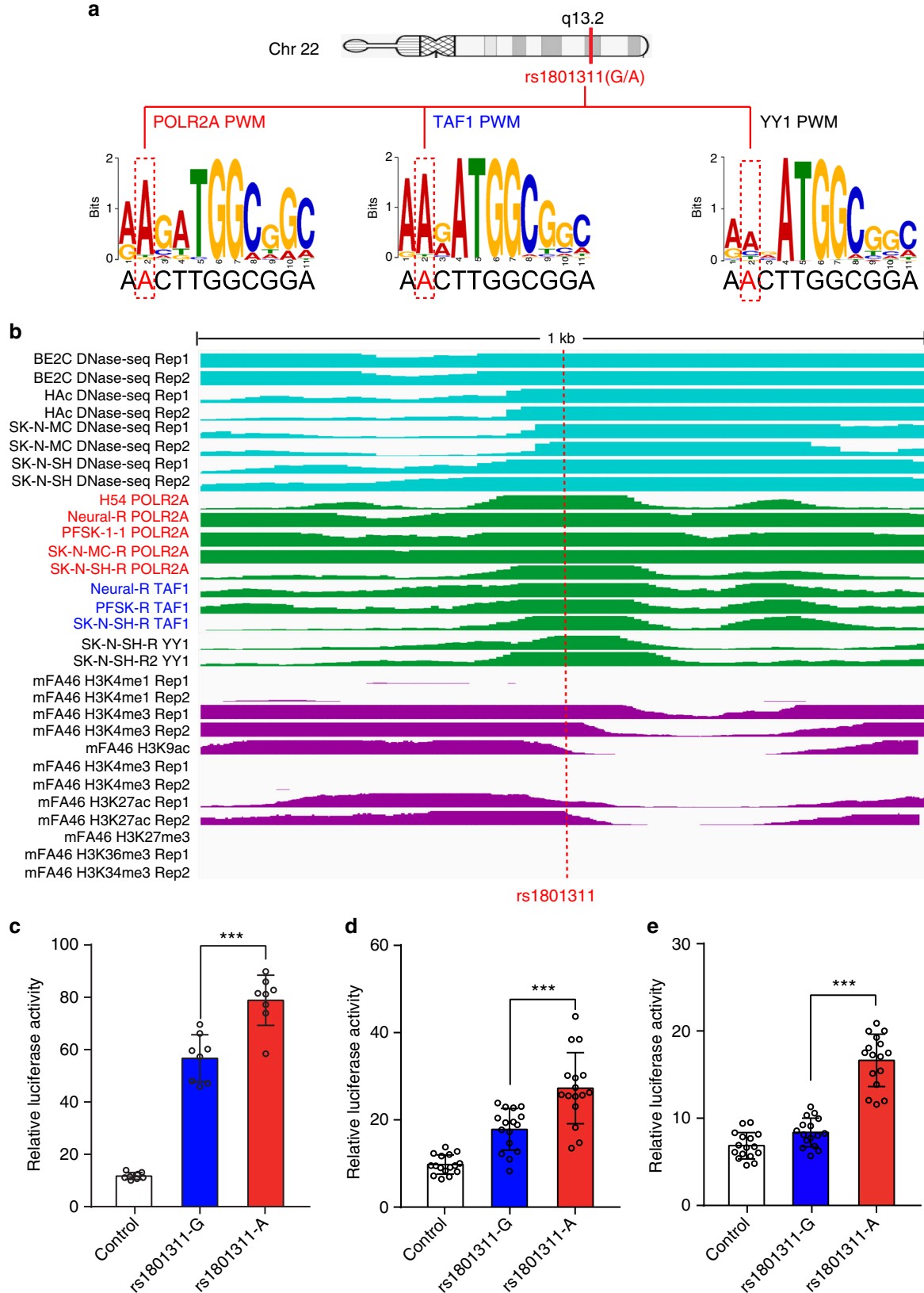

higher expression activity compared with A allele in HEK293 and SH-SY5Y cells (*P* < 0.001, two-tailed Student's *t*-test) (Fig. 5f, h). For rs7012106, the construct containing G allele exhibited significant higher luciferase activity compared with the construct containing the C allele in all three tested cell lines (*P* < 0.001, two-tailed Student's *t*-test) (Fig. 6b–d). The results of reporter gene assays provide further evidence that support the regulatory effect of these three CTCF binding–disrupting SNPs.

**Fig. 4** A single-nucleotide polymorphism (SNP; rs1801311) at schizophrenia risk locus 22q13.2 disrupts binding of three transcription factors (TFs). **a** SNP rs1801311 is located in the binding motif of POLR2A, TAF1 and YY1 TFs, and disrupts occupied POLR2A, TAF1 and YY1 binding site in human brain tissues or neuronal cell lines. Position weight matrices (PWMs) of corresponding TFs and the location of rs1801311 are shown (with red dotted box). **b** Genomic region (1 kb) surrounding SNP rs1801311 is shown with three featured visualization tracks, including DNase-Seq signal (light blue), chromatin immunoprecipitation and sequencing (ChIP-Seq) signal (green) for the selected TF and histone modifications (purple). The heights of the colored graphs (in **b**) reflect the ChIP-Seq and DNase-Seq signal intensities, which were scaled from 0 to 50 (signal intensities larger than 50 were truncated and not shown). The location of rs1801311 is highlighted with the dotted red line. **c–e** Reporter gene assays showed that the allelic differences at the rs1801311 influenced the luciferase activity significantly in HEK293 cells (**c**), SK-N-SH cells (**d**) and SH-SY5Y cells (**e**). Of note, the A allele of rs1801311 conferred a significant higher activity compared with G allele in all three tested cell lines. Data represent mean ± SD; $n = 8$ for each group in **c** HEK293 cells; $n = 16$ for each group in **d** SK-N-SH cells and **e** SH-SY5Y cells. ***$P < 0.001$, two-tailed Student's $t$-test. Source data are provided as a Source Data file

**Disruption of other TF binding by regulatory SNPs**. We demonstrated how a regulatory SNP (rs2270363, located in 16p13.3) disrupted binding of USF1 and MAX1 through integrating data from ChIP-Seq and PWM (Fig. 6e). SNP rs2270363 was located in upstream of *NMRAL1* and *HMOX2* genes (these two genes have opposite transcription direction). However, as *HMOX2* has multiple isoforms, rs2270363 is also located in the intron 1 of the longest transcript of *HMOX2*. DNase-Seq profiles and histone modification data showed that rs2270363 was located within an active transcription genomic region (Fig. 6e). In addition, ChIP-Seq profiles indicated that USF1 and MAX1 could bind to the region containing rs2270363 in human neuronal cell line. rs2270363 was located in binding motif of USF1 and MAX1, and FIMO analysis showed that rs2270363 disrupted occupied USF1 and MAX1 binding sites. We amplified the DNA fragments containing different alleles of rs2270363 and cloned them into pGL4.11 vector. We then compared the luciferase activity driven by the cloned DNA fragments containing different alleles of rs2270363. The construct containing A allele of rs2270363 exhibited significant higher luciferase activity compared with the construct containing the G allele in all three tested cell lines ($P < 0.01$, two-tailed Student's $t$-test) (Fig. 6f–h). In addition to rs2270363, USF1 binding was disrupted by other regulatory SNPs. For example, SNP rs7014953 also disrupted the binding of USF1 (Fig. 7a). We confirmed the regulatory effect of rs7014953 in SK-N-SH and SH-SY5Y cells ($P < 0.001$, two-tailed Student's $t$-test) (Fig. 7b, c).

In addition to above-mentioned TFs, the identified regulatory SNPs also disrupted bindings of other TFs. As shown in Fig. 7d, TCF12 binding was disrupted by SNP rs6992091. Again, we validated the regulatory effect of rs6992091 (Fig. 7e, f) with reporter gene assays. The construct containing G allele of rs6992091 exhibited a significantly higher luciferase activity compared with the construct containing the A allele in SK-N-SH and SH-SY5Y cells ($P < 0.001$, two-tailed Student's $t$-test). Taken together, these results provide further evidence that supports the identified TF binding–disrupting SNPs are regulatory SNPs.

**Regulatory effect of additional TF binding–disrupting SNPs**. We identified 132 SCZ-associated regulatory SNPs that disrupted bindings of TFs through annotating potential causal SNPs with ChIP-Seq and PWM data (Supplementary Data 3). These 132 TF binding–disrupting SNPs were from 81 SCZ risk loci (i.e., corresponding to 81 index SNPs). Most of the regulatory SNPs were located in intronic and intergenic regions (Fig. 3a). The wide overlapping of the regulatory SNPs with H3K27ac histone modification (http://www.szdb.org/) suggested that these SNPs were mainly located in enhancer regions. We thus performed reporter gene assays to validate the regulatory effect of some of the identified TF binding–disrupting SNPs. The selection criteria of TF binding–disrupting SNPs for reporter gene assays were as

follows. First, this TF binding–disrupting SNP was associated with expression of the same gene simultaneously in at least two independent eQTL datasets. Second, this SNP was located in a genomic region with strong DNase-Seq or Histone modification signal. In addition to the 7 SNPs investigated above (i.e., rs1801311, rs12912934, rs16937, rs7012106, rs2270363, rs7014953 and rs6992019) (Figs. 4–7), regulatory effects of two additional SNPs (rs2535629 and rs2711116) were also tested with reporter gene assays (Supplementary Figs 4 and 5). In total, we studied the regulatory effect of 9 SNPs and found that all of the cloned genomic sequences (containing the tested SNPs) enhanced the activity of luciferase compared with control vector (empty pGL3-promoter) in the tested cell lines (Figs. 4–7, Supplementary Figs 4 and 5), supporting that these SNPs were located in enhancer regions. We also compared the effect of different alleles on luciferase activity and found that all of the tested SNPs showed a significant difference in luciferase activity for the two given alleles in at least one of the tested cells (Figs. 4–7, Supplementary Figs 4 and 5). These allele-specific reporter gene assays indicated that genetic variations (i.e., allelic differences) at the regulatory SNPs influenced the luciferase activity significantly, providing further evidence that supports the regulatory effects of the identified SNPs.

**Identification of the target genes of the regulatory SNPs**. To further identify the potential target genes regulated by the 132 identified regulatory SNPs, we examined the association between the identified regulatory SNPs and gene expression in human brain tissues in three independent brain eQTL datasets, including CMC[35], LIBD eQTL browser[33] and GTEx[48]. Tissues from the prefrontal cortex were used in CMC and LIBD datasets. Tissues from 13 brain regions (sample size range from 80 to 154) (Supplementary Table 3) were included in GTEx. We utilized the default FDR or $P$ thresholds used in the original papers. In CMC dataset, the default FDR threshold is 0.05. In LIBD dataset, the default FDR is 0.01. In GTEx, we used $P < 0.001$ as the threshold. We found that 97 of the 132 regulatory SNPs were significantly associated with gene expression in human brain (FDR < 0.05 in CMC, FDR < 0.01 in LIBD and $P < 0.001$ in GTEx) (Supplementary Data 4). Among the 132 regulatory SNPs, 58 showed significant association with the expression of the same gene in at least two independent eQTL datasets (Supplementary Table 5), suggesting these genes are true targets of the identified regulatory SNPs. Nevertheless, further functional assays (e.g., reporter gene assays) are needed to validate if these genes represent true targets of the identified regulatory SNPs. Of note, 29 regulatory SNPs showed significant association with the expression of the same gene in three independent brain eQTL datasets (e.g., rs1801311 is significantly associated with *FAM109B* expression in all of the three eQTL datasets) (Table 1), strongly suggesting that these genes were regulated by these regulatory SNPs. Interestingly, we noticed that six distinct regulatory SNPs (rs1801311, rs2269524,

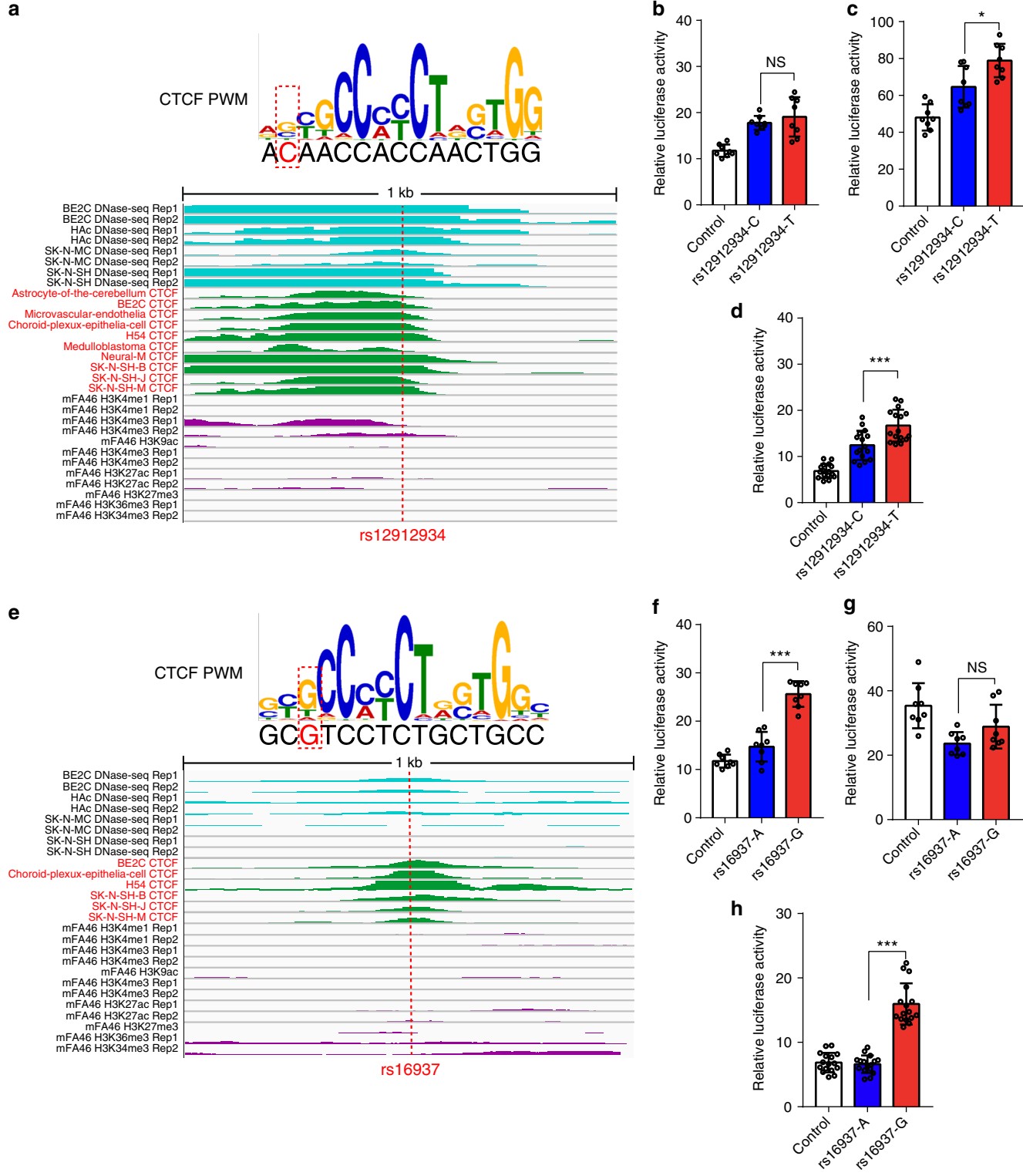

**Fig. 5** Disruption of CTCF binding by risk single-nucleotide polymorphisms (SNPs; rs12912934 and rs16937). **a** SNP rs12912934 is located in CTCF binding motif and disrupts occupied CTCF binding site. The matched position weight matrix (PWM) and the location of rs12912934 (red dotted line) are shown. DNase-Seq and chromatin immunoprecipitation and sequencing (ChIP-Seq) experiments showed that rs12912934 is located in an actively transcribed region, with strong DNase-Seq and ChIP-Seq signals. **b–d** Reporter gene assays revealed that the allelic differences at the rs12912934 affected the luciferase activity significantly in SK-N-SH (**c**) and SH-SY5Y cells (**d**). **e** SNP rs16937 is located in CTCF binding motif, a region with strong ChIP-Seq signals in brain tissues or neuronal cell lines. **f–h** Reporter gene assays showed that the allelic differences at the rs16937 influenced the luciferase activity significantly in HEK293 (**f**) and SH-SY5Y cells (**h**). Data represent mean ± SD; $n = 8$ for each group in **b**, **f** (HEK293 cells) and **c**, **g** (SK-N-SH cells); $n = 16$ for each group in **d**, **h** (SH-SY5Y cells). NS, not significant, *$P < 0.05$, ***$P < 0.001$, two-tailed Student's $t$-test. Source data are provided as a Source Data file

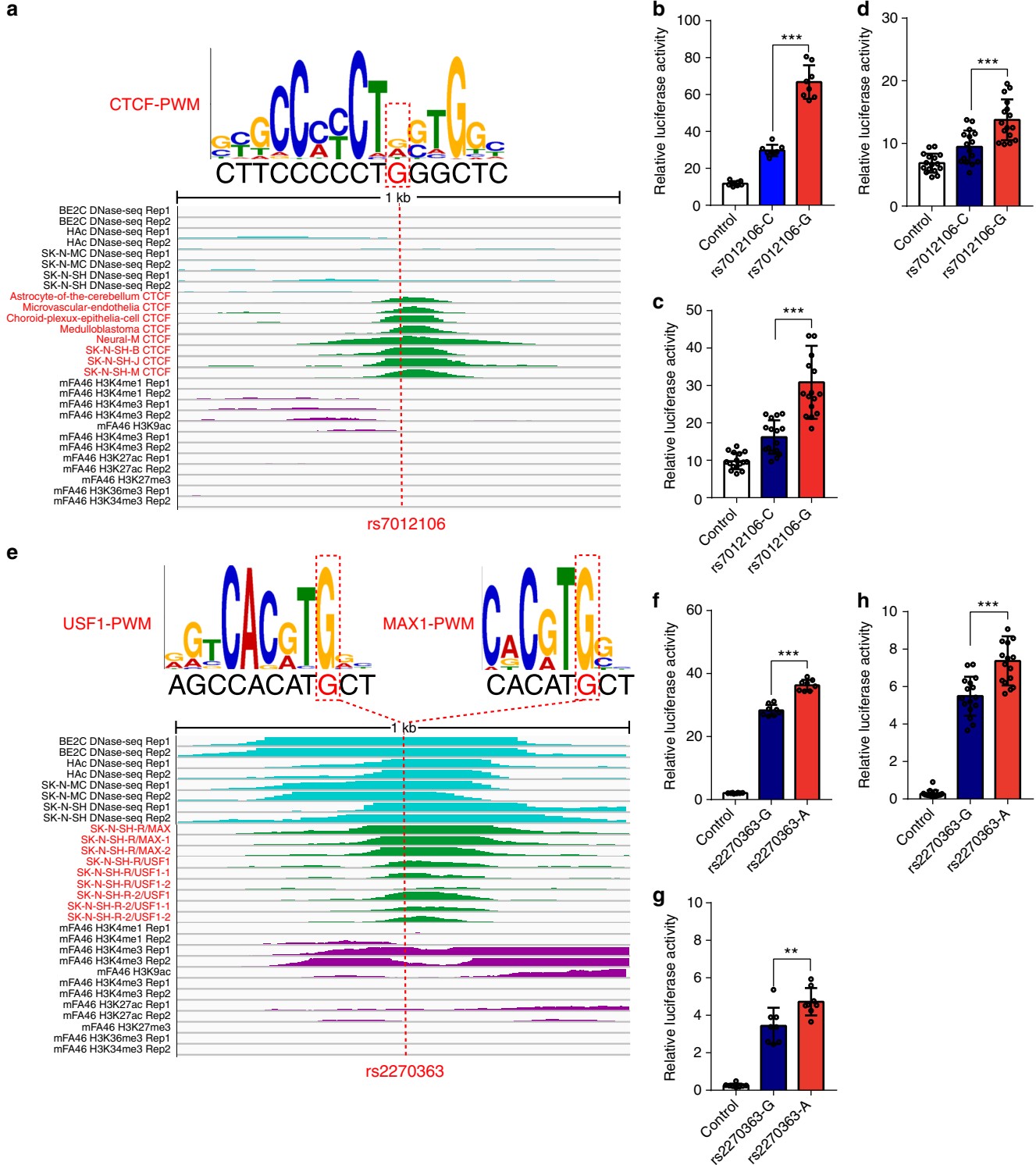

**Fig. 6** Disruption of CTCF, USF1 and MAX1 binding by schizophrenia risk single-nucleotide polymorphisms (SNPs; rs7012106 and rs2270363). **a** SNP rs7012106 is located in CTCF binding motif and disrupts occupied CTCF binding site. **b–d** Reporter gene assays showed that the construct containing G allele exhibited significant higher luciferase activity compared with the construct containing the C allele in all three tested cell lines (**b–d**). **e** Disruption of USF1 and MAX1 binding by SNP rs2270363. SNP rs2270363 is located in binding motif of USF1 and MAX1 and disrupts occupied USF1 and MAX1 binding site. Genomic region (1 kb) surrounding SNP rs2270363 is shown with three featured visualization tracks, including DNase-Seq signal (light blue), chromatin immunoprecipitation and sequencing (ChIP-Seq) signal (green) for the selected transcription factor (TF) and histone modifications (purple). **b–d** Reporter gene assay showed that the construct containing A allele (of rs2270363) exhibited significant higher luciferase activity compared with the construct containing the G allele in all three tested cell lines (**f–h**). Data represent mean ± SD; $n = 8$ for each group in **b**, **f** (HEK293 cells) and **g** (SK-N-SH cells); $n = 16$ for each group in **c** (SK-N-SH cells) **d**, **h** (SH-SY5Y cells). **$P < 0.01$, ***$P < 0.001$, two-tailed Student's $t$-test. Source data are provided as a Source Data file

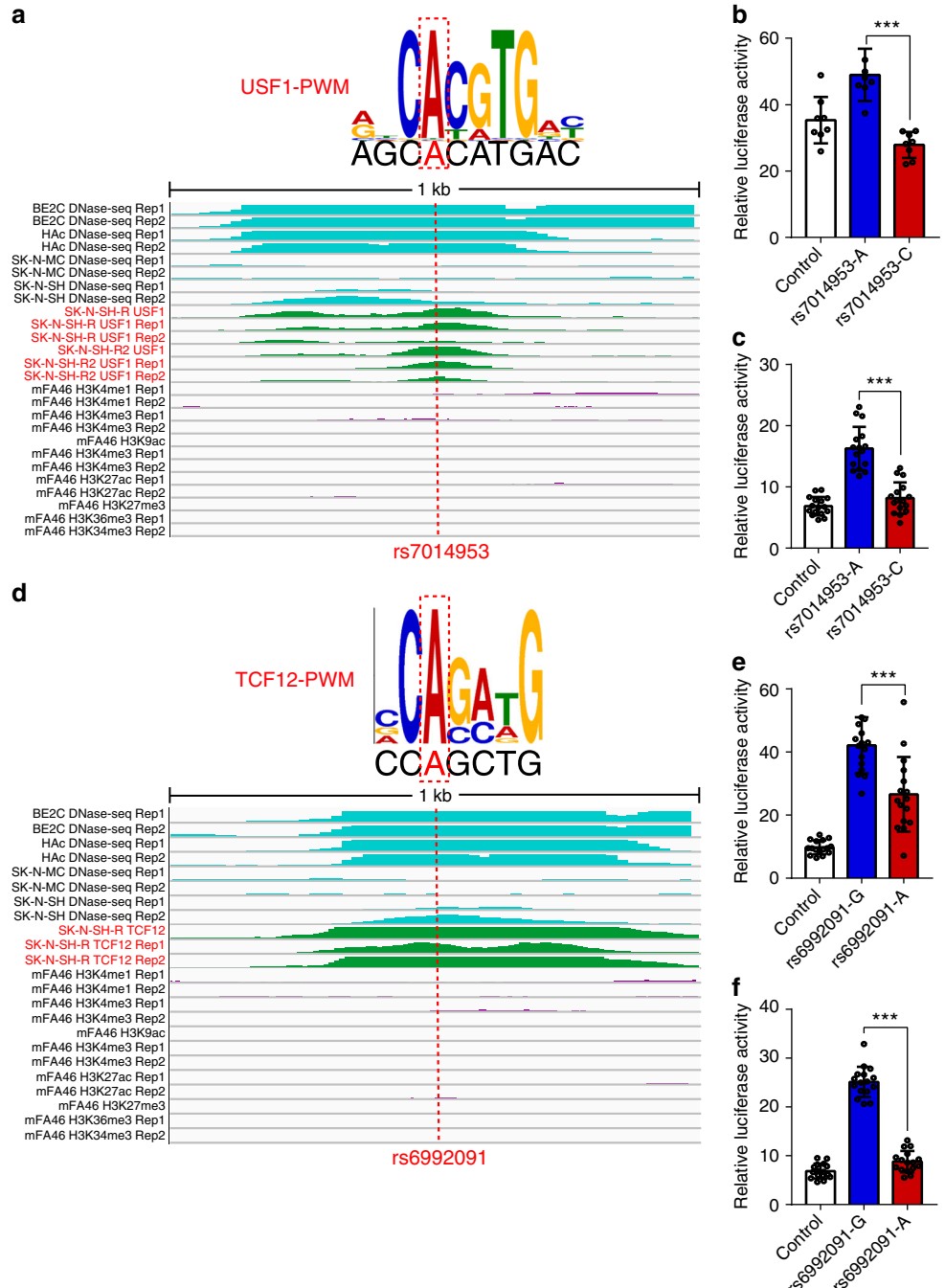

**Fig. 7** Disruption of USF1 and TCF12 binding by risk single-nucleotide polymorphisms (SNPs) rs7014953 and rs6992091. **a** SNP rs7014953 is located in binding motif of USF1 and disrupts occupied USF1 binding site. The matched position weight matrix (PWM) and the location of rs7014953 (red dotted line) are shown. DNase-Seq and chromatin immunoprecipitation and sequencing (ChIP-Seq) experiments showed that rs12912934 overlaps with DNase-Seq and ChIP-Seq peaks, indicating that USF1 can bind to the genomic region containing rs12912934 in human brain tissues or neuronal cell lines. **b**, **c** Reporter gene assay showed that the construct containing A allele (of rs7014953) exhibited significant higher luciferase activity compared with the construct containing the C allele in SK-N-SH (**b**) and SH-SY5Y (**c**) cells. **d** SNP rs6992091 is located in the binding motif of TCF12 and disrupts occupied TCF12 binding site in human brain tissues or neuronal cell lines. Matched PWM of TCF12 and the location of rs6992091 are shown (with red dotted box). Genomic region (1 kb) surrounding SNP rs6992091 is shown with three featured visualization tracks, including DNase-Seq signal (light blue), ChIP-Deq signal (green) for TCF12 and histone modifications (purple). The location of rs6992091 is highlighted with the dotted red line. **e**, **f** Reporter gene assay showed that the construct containing G allele (of rs6992091) exhibited significant higher luciferase activity compared with the construct containing the A allele in SK-N-SH (**b**) and SH-SY5Y (**c**) cells. Data represent mean ± SD; $n = 8$ for each group in **b** (SK-N-SH cells) and $n = 16$ for each group in **c**, **f** (SH-SY5Y cells) and **e** (SK-N-SH cells). ***$P < 0.001$, two-tailed Student's $t$-test. Source data are provided as a Source Data file

**Table 1 Association significance between the TF binding–disrupting SNPs and gene expression in human brain tissues**

| SNP | Gene | P (CMC) | FDR (CMC) | P (LIBD) | FDR (LIBD) | P (GTEx) |
|---|---|---|---|---|---|---|
| rs10795 | GOLGA2P7 | 1.63E−17 | 6.37E−15 | 1.07E−09 | 1.88E−07 | 4.50E−13 (Cau) |
| rs12136320 | TMEM81 | 2.97E−05 | 2.00E−03 | 9.91E−06 | 7.98E−04 | 1.10E−06 (Putamen) |
| rs12912934 | GOLGA2P7 | 2.75E−28 | 2.91E−25 | 9.30E−19 | 4.52E−16 | 1.20E−24 (Cau) |
|  | EFTUD1P1 | 4.07E−13 | 9.63E−11 | 8.48E−07 | 8.84E−05 | 1.10E−05 (Hyp) |
|  | DNM1P51 | 9.84E−05 | 4.96E−03 | 5.54E−10 | 1.02E−07 | 2.50E−11 (Front Cor) |
| rs16937 | TMEM81 | 1.06E−05 | 8.06E−04 | 3.64E−10 | 6.88E−08 | 9.70E−07 (Putamen) |
|  | RBBP5 | 1.16E−09 | 1.84E−07 | 1.53E−04 | 8.56E−03 | 2.70E−09 (NAc) |
| rs1801311 | NAGA | 4.06E−17 | 1.43E−14 | 4.46E−29 | 4.42E−26 | 5.70E−10 (Cere) |
|  | FAM109B | 1.46E−06 | 1.34E−04 | 6.23E−17 | 2.58E−14 | 6.30E−06 (Cortex) |
| rs223390 | LRRC37A15P | 2.58E−13 | 4.55E−11 | 1.43E−04 | 8.10E−03 | 2.30E−09 (Cere Hemi) |
| rs2269524 | NAGA | 5.33E−18 | 2.03E−15 | 1.04E−26 | 8.81E−24 | 2.90E−09 (Cere) |
|  | FAM109B | 1.02E−06 | 9.73E−05 | 7.35E−16 | 2.77E−13 | 5.80E−06 (Cortex) |
| rs2270363 | CORO7 | 2.06E−10 | 5.06E−08 | 1.72E−17 | 7.46E−15 | 8.90E−13 (ACC) |
|  | NMRAL1 | 1.62E−10 | 4.05E−08 | 4.88E−16 | 1.87E−13 | 9.00E−41 (Cere Hemi) |
|  | CDIP1 | 3.05E−07 | 4.40E−05 | 1.43E−05 | 1.11E−03 | 5.10E−34 (Cere) |
| rs2304204 | CPT1C | 8.51E−14 | 2.55E−11 | 4.56E−05 | 3.05E−03 | 2.00E−05 (Cere) |
| rs2385395 | CNPPD1 | 4.21E−06 | 3.11E−04 | 1.10E−06 | 1.12E−04 | 4.10E−11 (Spinal cord) |
| rs2711116 | DFNA5 | 5.07E−04 | 1.88E−02 | 6.63E−08 | 8.62E−06 | 2.00E−07 (Cortex) |
| rs281759 | FTCDNL1 | 2.98E−13 | 6.79E−11 | 3.92E−11 | 8.51E−09 | 2.00E−06 (Hyp) |
|  | TYW5 | 2.25E−09 | 3.07E−07 | 1.80E−06 | 1.74E−04 | 5.60E−09 (Cere Hemi) |
| rs2856268 | RLBP1 | 2.58E−12 | 5.56E−10 | 1.28E−08 | 1.89E−06 | 6.50E−06 (Cortex) |
|  | POLG | 5.20E−05 | 2.87E−03 | 4.47E−06 | 3.93E−04 | 7.20E−09 (Cere) |
| rs2974999 | TOM1L2 | 6.61E−14 | 4.75E−12 | 1.20E−04 | 6.97E−03 | 2.90E−06 (Cere) |
| rs340836 | PROX1−AS1 | 9.71E−06 | 7.46E−04 | 3.16E−07 | 3.60E−05 | 3.30E−07 (Cere) |
| rs3814880 | INO80E | 2.28E−13 | 7.83E−11 | 7.60E−29 | 7.44E−26 | 3.30E−10 (Cere) |
| rs3822346 | PCDHA10 | 1.96E−14 | 4.50E−12 | 1.30E−13 | 3.86E−11 | 1.40E−11 (Front Cor) |
|  | PCDHA13 | 3.90E−19 | 1.49E−16 | 3.21E−14 | 1.02E−11 | 4.90E−10 (Cere Hemi) |
|  | PCDHA8 | 2.43E−18 | 8.29E−16 | 5.03E−32 | 5.90E−29 | 3.20E−07 (Cere Hemi) |
|  | PCDHA2 | 3.69E−08 | 3.96E−06 | 3.96E−06 | 3.53E−04 | 6.80E−08 (Cere) |
|  | PCDHA7 | 9.78E−20 | 4.04E−17 | 1.30E−14 | 4.29E−12 | 5.80E−09 (Spinal cord) |
| rs4786494 | CORO7 | 2.78E−10 | 6.71E−08 | 2.57E−20 | 1.42E−17 | 3.10E−13 (ACC) |
| rs5751195 | NAGA | 6.64E−19 | 2.71E−16 | 1.49E−23 | 1.03E−20 | 1.80E−21 (Cere) |
|  | WBP2NL | 2.05E−41 | 4.08E−38 | 7.53E−12 | 1.81E−09 | 2.70E−09 (ACC) |
| rs6002621 | FAM109B | 3.09E−06 | 2.67E−04 | 3.53E−15 | 1.24E−12 | 3.40E−06 (Cortex) |
|  | NAGA | 2.19E−17 | 7.87E−15 | 1.87E−23 | 1.29E−20 | 2.60E−09 (Cere) |
| rs61660810 | USP32P3 | 5.82E−10 | 3.21E−08 | 1.40E−04 | 7.96E−03 | 2.00E−06 (Cortex) |
| rs6992091 | FAM86B3P | 5.79E−29 | 5.69E−26 | 7.93E−29 | 7.76E−26 | 7.30E−10 (Front Cor) |
| rs7014953 | FAM86B3P | 2.19E−26 | 2.19E−26 | 1.01E−26 | 8.58E−24 | 3.10E−09 (Front Cor) |
| rs72748702 | EFTUD1P1 | 7.36E−14 | 1.95E−11 | 2.07E−06 | 1.97E−04 | 1.40E−05 (Hyp) |
|  | DNM1P51 | 1.33E−04 | 6.41E−03 | 5.39E−09 | 8.46E−07 | 2.20E−11 (Front Cor) |
|  | GOLGA2P7 | 1.40E−27 | 1.40E−24 | 2.36E−18 | 1.11E−15 | 1.90E−24(Cau) |
| rs778593 | PCDHA10 | 5.19E−10 | 7.48E−08 | 1.50E−12 | 3.93E−10 | 2.40E−08 (Front Cor) |
|  | PCDHA13 | 1.06E−12 | 2.08E−10 | 2.60E−10 | 5.03E−08 | 4.00E−06 (Cere) |
|  | PCDHA8 | 4.71E−14 | 1.04E−11 | 6.25E−21 | 3.61E−18 | 9.10E−06 (Cere Hemi) |
|  | ZMAT2 | 6.87E−10 | 9.75E−08 | 1.28E−08 | 1.90E−06 | 2.40E−06 (ACC) |
|  | WDR55 | 1.27E−04 | 6.37E−03 | 3.83E−05 | 2.62E−03 | 1.90E−06 (Cere) |
| rs78532287 | USP32P3 | 4.48E−10 | 2.49E−08 | 9.14E−13 | 2.46E−10 | 5.60E−06 (NAc) |
|  | CCDC144C | 6.20E−17 | 5.16E−15 | 2.90E−11 | 6.42E−09 | 1.70E−06 (Putamen) |
| rs796364 | FTCDNL1 | 1.01E−13 | 2.46E−11 | 9.50E−09 | 1.44E−06 | 5.80E−07 (Hyp) |
|  | TYW5 | 1.28E−08 | 1.54E−06 | 1.99E−06 | 1.91E−04 | 2.50E−08 (Cere Hemi) |
| rs8135801 | NAGA | 1.14E−17 | 4.22E−15 | 6.67E−29 | 6.55E−26 | 5.10E−09 (Cere) |
|  | FAM109B | 6.60E−07 | 6.55E−05 | 3.84E−17 | 1.62E−14 | 6.30E−06 (Cortex) |
| rs9362397 | C6orf162 | 3.16E−21 | 1.01E−18 | 4.47E−25 | 3.42E−22 | 7.70E−17 (Cere) |

Brain tissues from the dorsolateral prefrontal cortex (DLPFC) were used in CMC (N = 467) and LIBD brain eQTL browser (N = 412). Tissues from 8 brain regions were used in GTEx. Only SNPs that showed significant association with the expression of the same gene in all three independent expression quantitative locus (eQTL) datasets are shown
*TF* transcription factor, *SNP* single-nucleotide polymorphism, *CMC* CommonMind Consortium, *FDR* false discovery rate, *LIBD* Lieber Institute for Brain Development, *GTEx* Genotype-Tissue Expression, *Front Cor* frontal cortex, *Cau* caudate, *ACC* anterior cingulate cortex, *NAc* nucleus accumbens, *Cere* cerebellum, *Hyp* hypothalamus, *Cere* Hemi cerebellar hemisphere, *Sub nig* substantia nigra, *NA* not available

rs28633410, rs5751195, rs6002621 and rs8135801) showed significant association with the expression of *FAM109B* gene (Supplementary Table 5 and Table 1), indicating that these SNPs may confer risk of schizophrenia through modulating *FAM109B* expression. In addition to *FAM109B*, expression of other genes (including *TMEM81, EFTUD1P1, BDH2, RERE, CORO7, NAGA,* etc) also showed significant association with different regulatory

SNPs. Considering that these regulatory SNPs are located in DNA binding motifs (with corresponding PWM match) and allelic differences at these SNPs disrupt binding sites of TFs, the significant associations between the regulatory SNPs and genes in independent brain eQTL datasets strongly suggest that these genes are true targets for the identified regulatory SNPs. Further functional experiments (such as reporter gene assays and

CRISPR/CAS9 (clustered regularly interspaced short palindromic repeats/CRISPR-associated 9) experiments) are needed to verify if these genes are regulated by these regulatory SNPs.

To identify all of the possible target genes of the regulatory SNPs, we performed eQTL annotation (using CMC dataset) to test the associations of the regulatory SNPs with all genes in close physical proximity (e.g., ±2 Mb). For each regulatory SNP (the TF binding–disrupting SNPs are listed in Supplementary Data 3, a total of 132 SNPs), we performed eQTL annotation through using the genotype and expression data from the CMC dataset. The eQTL analysis was conducted with PLINK (v1.9). The associations of the regulatory SNPs with all genes in close physical proximity (e.g., ±2 Mb) are listed in Supplementary Data 5 (Of note, only associations with a $P < 0.01$ are shown).

To explore if the identified regulatory SNPs were significantly associated with gene expression than random SNPs, we conducted 10,000 simulations (Methods). Compared with the random SNPs, the identified regulatory SNPs showed significant association with gene expression ($Z = 14.23$, $P = 3.09 \times 10^{-46}$, Methods) (Supplementary Fig. 6), further supporting the potential regulatory effect of the identified TF binding–disrupting SNPs. Collectively, these results identified the potential target genes of the identified TF binding–disrupting SNPs and suggested that these regulatory SNPs may confer schizophrenia risk through affecting the expression of their target genes.

**Linkage disequilibrium analysis between the regulatory SNPs.** In some cases, we identified several TF binding–disrupting SNPs at a single schizophrenia risk locus. For example, we identified 6 TF binding–disrupting SNPs (rs1801311, rs2269524, rs28633410, rs5751195, rs6002621 and rs8135801) at 22q13.2. To investigate the LD between the 132 TF binding–disrupting SNPs, we performed LD analysis using SNiPA (a tool for annotating and browsing genetic variants)[49]. We found that 40 TF binding–disrupting SNPs showed LD ($r^2 > 0.3$) with other TF binding–disrupting SNPs in Europeans (Supplementary Table 6). Of note, 8 TF binding–disrupting SNPs (rs10083370, rs2304206, rs3773744, rs4759413, rs60754073, rs76514049, rs78866909 and rs9616378) showed complete LD (i.e., $r^2 = 1$) with other TF binding–disrupting SNPs (Supplementary Table 6). These LD results indicate that some of the TF binding–disrupting were in linkage disequilibrium with each other. However, considering that all of these 132 SNPs disrupt binding of TFs, suggesting that these 132 TF binding–disrupting SNPs may have functional consequences. Thus, more work is needed to investigate if all of the highly linked TF binding–disrupting SNPs or only some of the highly linked TF binding–disrupting SNPs confer risk of schizophrenia through affecting gene expression.

**Allele-specific expression of the regulatory SNPs.** Our eQTL annotation showed that most of the identified regulatory SNPs were associated with gene expression in brain tissues. To further verify the *cis* regulatory effects of the identified TF binding–disrupting SNPs, we extracted the allele-specific expression (ASE) results of the identified TF binding–disrupting SNPs from GTEx[50]. Of note, only very limited regulatory SNPs were suitable for ASE analysis (as ASE analysis requires that the analyzed SNP is heterozygous and expressed in the same individual). We found that 10 TF binding–disrupting SNPs showed significant ($P < 0.001$, Binomial test) allelic imbalance (i.e., allele-specific expression) in human brain tissues (Fig. 8a–j). For example, the C allele of SNP rs1321 was preferentially expressed compared with the T allele ($P = 2.84 \times 10^{-4}$) (Fig. 8a), and G allele of SNP rs16937 was preferentially expressed compared with A allele ($P = 8.19 \times 10^{-5}$) (Fig. 8b). Of note, reporter gene assays

also showed that the G allele of rs16937 conferred a significant higher expression activity compared with A allele in HEK293 and SH-SY5Y cells ($P < 0.01$). These ASE results provide further evidence that supports the regulatory effects of the identified TF binding–disrupting SNPs.

**Pathway analysis of the target genes.** To explore if the target genes (Supplementary Data 4) of the regulatory SNPs were enriched in specific pathways, we carried out gene ontology (GO) analysis using DAVID (Database for Annotation, Visualization and Integrated Discovery)[51]. We found that two categories (including "cell adhesion" and "nervous system development") were significantly enriched ($P < 0.05$, corrected by the Benjamini–Hochberg procedure) among the target genes of the regulatory SNPs (Supplementary Table 7). The over-representation of nervous system development genes provides further evidence that supports the neurodevelopmental hypothesis of schizophrenia[40].

**Spatio-temporal expression pattern of target genes.** We carried out spatio-temporal expression pattern analysis using expression data from the BrainSpan (http://www.brainspan.org/). Two groups of target genes (i.e., genes listed in Supplementary Data 4 and Supplementary Table 5) were used for spatio-temporal expression pattern analysis. Genes whose expression were associated with regulatory SNPs in any eQTL dataset were listed in Supplementary Data 4. However, genes whose expression are associated with regulatory SNPs in at least two eQTL datasets are listed in Supplementary Table 5. Thus, genes listed in Supplementary Table 5 are a subset of genes listed in Supplementary Data 4. Spatio-temporal expression pattern analysis was carried out as previously described[52]. We found that the expression level of the target genes in Supplementary Table 5 was significantly higher than the background genes across all developmental stages ($P < 1.0 \times 10^{-7}$, Wilcoxon rank sum test). Similarly, the expression level of the target genes in Supplementary Data 4 was also significantly higher than the background genes across all developmental stages ($P < 5.0 \times 10^{-7}$) (Fig. 8k). In addition, we also found that the expression level of the target genes was higher in prenatal stage compared with postnatal stage ($P = 0.001$ for gene set in Supplementary Table 5 and $P = 0.008$ for gene set in Supplementary Data 4) (Fig. 8k), suggesting that the target genes may play a role in brain development.

**Cell type-specific expression analysis of target genes.** To explore the expression pattern of the target genes of the identified regulatory SNPs in different brain cell types, we extracted the specificity value of each target gene in 24 cell types defined by Skene et al.[53]. We found that three cell types contained more than 6 target genes at the specificity cutoff of 0.1 (i.e., specificity score >0.1), with pyramidal cells having the most number of target genes (Fig. 8l). This result is consistent with previous findings[53] and provides further support for the involvement of pyramidal cells in schizophrenia.

In conclusion, we used two complementary approaches (functional annotation and functional genomics) to systematically annotate the GWAS significant schizophrenia risk loci (Supplementary Fig. 7). The functional annotation prioritized 153 overlapping top functional SNPs and 66 of the 153 prioritized functional SNPs were associated with gene expression in human brains. Reporter gene assays were conducted for 10 prioritized functional SNPs and 7 SNPs showed significant allelic effects on reporter gene activity. In addition, we derived binding motifs of 30 TFs through integrating ChIP-Seq experiments from brain tissues (or neuronal cells) and PWMs. We mapped the potential

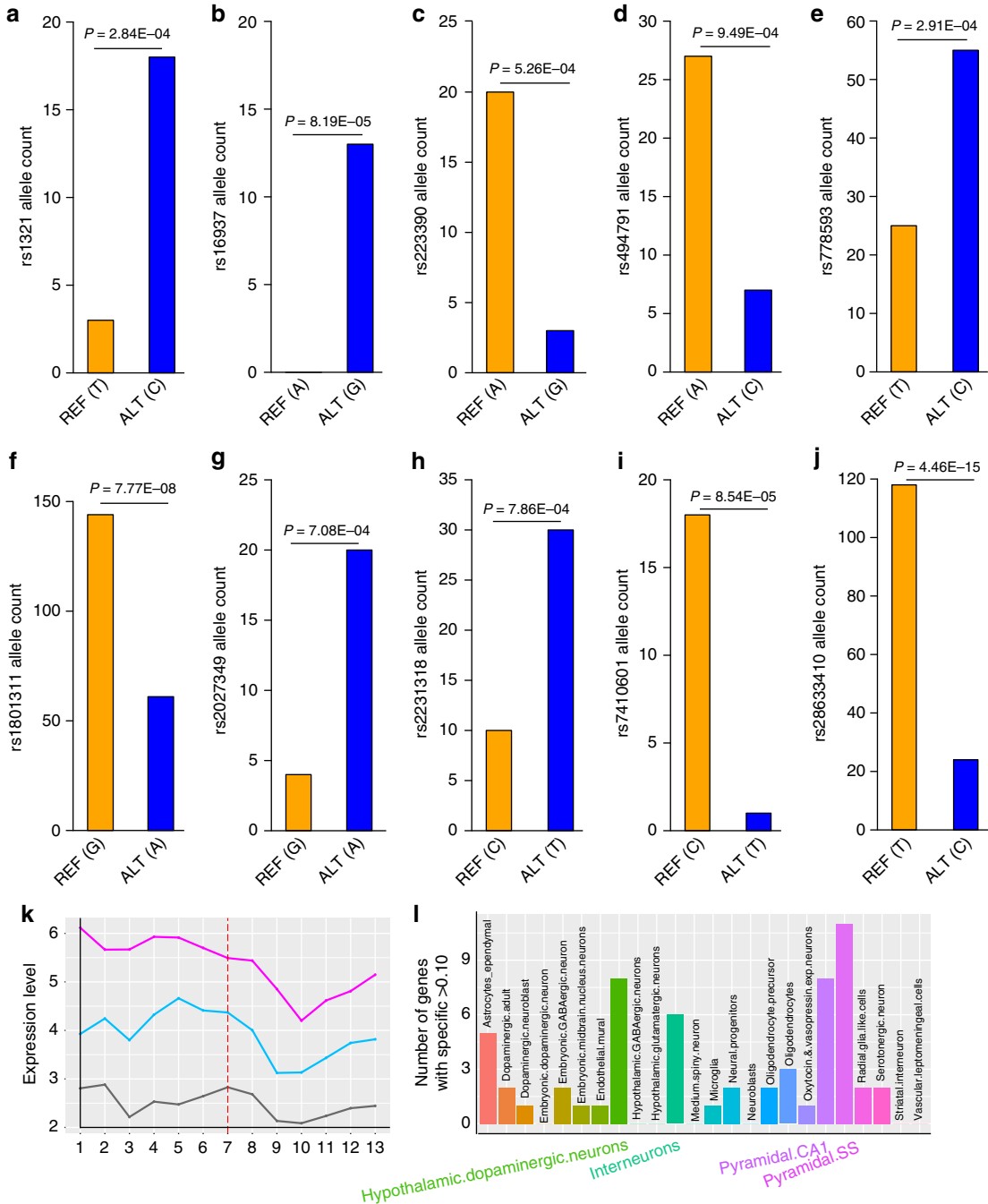

**Fig. 8** Allele-specific expression analysis results and expression pattern of target genes. **a–j** Ten transcription factor (TF) binding–disrupting single-nucleotide polymorphisms (SNPs) showed significant (Binomial $P < 0.001$) allele-specific expression (allelic imbalance) in human brain tissues from the GTEx. For each SNP, the counts of the reference allele and the alternative allele are shown. Of note, reporter gene assays also support that different alleles at rs1801311 and rs16937 conferred significant differences in luciferase activity, further supporting the regulatory effect of rs1801311 and rs16937. **k** Expression pattern of target genes of the identified regulatory SNPs across different developmental stages. The target genes of the identified regulatory SNPs showed a significant higher expression level compared with background genes ($P < 1.0 \times 10^{-7}$ for gene set in Supplementary Table 5, and $P < 5.0 \times 10^{-7}$ for gene set in Supplementary Data 4). In addition, we also found that the expression level of the target genes was higher in prenatal stage compared with postnatal stage ($P = 0.001$ for gene set in Supplementary Table 5, and $P = 0.008$ for gene set in Supplementary Data 4). The gray line represents the expression level of the background genes, the pink line represents the expression level of target genes listed in Supplementary Table 5, and the light blue line represents the expression level of target genes listed in Supplementary Data 4. **l** Cell type-specific expression analysis of the target genes of the identified regulatory SNPs. Target genes with a specificity score >0.1 were extracted and the number of targets genes with a specificity score >0.1 were plotted. Of note, the pyramidal cells have the most number of target genes, suggesting that the expression of target genes were enriched in pyramidal cells. Source data are provided as a Source Data file

causal risk SNPs (a total of 23,400) from three schizophrenia GWASs to the derived binding motifs and identified 132 SNPs (from 81 schizophrenia risk loci) that disrupted binding of 21 distinct TFs. We found that 97 out of the 132 TF binding–disrupting SNPs were significantly associated with gene expression in human brain tissues. Finally, regulatory effects of 9 TF binding–disrupting SNPs were validated by reporter gene assays (Supplementary Fig. 7). Our study identified the plausible causal variants for schizophrenia and revealed the gene regulatory mechanisms affected by schizophrenia risk SNPs (including widespread disruption of POLR2A and CTCF binding). In addition, our study also identified the potential target genes of the TF binding–disrupting risk SNPs. It is likely that the identified TF binding–disrupting risk SNPs exert their effects on schizophrenia through modulating the expression level of these target genes.

## Discussion

In the past decade, significant advances have been made in understanding the genetic basis of schizophrenia. Since O'Donovan et al.[2] reported the first genome-wide significant risk locus (near *ZNF804A*) for schizophrenia in 2008, numerous novel risk loci have been identified. Despite the fact that over 180 schizophrenia risk loci have been identified by GWAS[12–14], pinpointing the causal (or functional) variants at the risk loci and elucidating their roles in schizophrenia susceptibility remain major challenges in psychiatric genetics. In this study, we conducted systematic and deep analyses to identify the potential causal variants and to investigate how these causal variants confer schizophrenia risk. We first prioritized the most possible functional SNP at each risk locus by using the well-characterized functional annotation approaches. We then identified 132 risk SNPs that disrupted binding of 21 distinct TFs through integrating Chip-Seq and PWMs. We validated the regulatory effects of some identified functional SNPs with reporter gene assays and allele-specific expression analysis. We further identified the potential target genes of the regulatory SNPs. Finally, we showed that nervous system development genes were significantly enriched among the target genes of the identified regulatory SNPs.

Our study has important implications for elucidating the genetic mechanisms of schizophrenia. First, through integrating the diverse data sources from functional genomics, we translated the association findings from schizophrenia GWAS into genetic and gene regulatory mechanisms. With the rapid increase in sample size, novel schizophrenia risk variants have been identified at the unprecedented rate. By contrast, identification of causal (or functional) variants and genes lags far behind. Our post-GWAS functional genomics analyses linked the identified risk loci with specific variants and genes, thus providing a starting point for further mechanistic and functional investigation. Second, we systematically identified the causal (or functional) variants and investigated the potential gene regulatory mechanisms at all of the genome-wide significant risk loci. Though previous studies have identified some causal variants (or genes) at several risk loci[19,54], large-scale identification of causal variants and characterization of gene regulation at the reported risk loci remain major challenges. Through analyzing the functional genomics data with bioinformatic methods, for the first time, we conducted a systematic and deep analysis to investigate the gene regulatory mechanisms at all of the reported schizophrenia risk loci. Third, our study identified 132 TF binding–disrupting SNPs. These SNPs disrupt binding sites of multiple TFs, thus representing the most plausible causal SNPs. We furthermore verified the regulatory effects of some TF binding–disrupting SNPs using reporter gene assays and allele-specific expression analysis. Fourth, our study revealed new gene

regulatory mechanisms affected by schizophrenia risk SNPs, including widespread disruption of POLR2A and CTCF binding. Fifth, we identified the potential target genes of the TF binding–disrupting SNPs through eQTL annotation (boxplots of brain eQTLs results (only SNPs listed in Supplementary Table 5 were plotted) are shown in Supplementary Data 6). It is likely that the TF binding–disrupting SNPs confer schizophrenia risk through modulating the expression of these target genes. Finally, this study provides an example to illuminate the gene regulatory mechanisms underpinning other psychiatric disorders (such as major depressive disorder and bipolar disorder). Most of risk variants identified by GWAS are located in non-coding regions, suggesting the identified risk variants confer schizophrenia risk through modulating gene expression. Currently, pinpointing the causal variant remains a daunting task as each risk locus usually contains multiple variants that show similar association significance (due to LD). By annotating all of the variants at each locus with functional genomics data, our study provides an example for identifying potential causal variants. This method can be applied to other psychiatric disorders.

Of note, we used two complementary methods (i.e., functional annotation and functional genomics, see Fig. 1) to prioritize or identify the potential functional (or causal) variants at the reported schizophrenia risk loci. We compared the potential functional SNPs identified by these two approaches (i.e., functional annotation and functional genomics) and found 6 overlapping functional SNPs (Supplementary Table 8 and Supplementary Fig. 8). The reasons why only 6 overlapping functional SNPs were identified by these two approaches are discussed in the Supplementary Discussion.

We compared the target genes of the identified TF binding–disrupting SNPs with previous findings from Gusev et al.[55] and found 44 overlapping genes (Supplementary Table 9 and Supplementary Fig. 9). These overlapping genes may represent promising candidates for schizophrenia as these genes were prioritized by two different methods. In addition, the significant enrichment of nervous system development-related genes among the target genes of the TF binding–disrupting SNPs provides further support for the neurodevelopmental hypothesis of schizophrenia[39,40,56]. More detailed discussion about the neurodevelopmental hypothesis of schizophrenia is provided in the Supplementary Discussion.

We noticed that CTCF binding was frequently disrupted by the schizophrenia risk SNPs, suggesting that disruption of CTCF binding may represent a common mechanism that schizophrenia risk SNPs exert their effect on schizophrenia. Interestingly, previous genetic studies also suggested that *CTCF* may have a role in schizophrenia[57,58]. More detailed information about the potential role of CTCF in SCZ is provided in Supplementary Discussion.

Binding of CTCF and POLR2A were frequently disrupted by schizophrenia risk SNPs. This may be due to the possibility that CTCF and POLR2A have more binding sites (on the genome) than other TFs. To test if disruption of CTCF and POLR2A were random, we conducted additional analyses. We found that the number of POLR2A binding–disrupting SNPs observed in this study were significantly higher than random SNPs ($P = 0.013$, 1000 simulations, Supplementary Fig. 10a), suggesting that the frequent disruption of POLR2A binding by schizophrenia risk SNPs may not be due to random effect. However, the number of CTCF binding–disrupting SNPs (observed in this study) were not significantly different from random SNPs ($P > 0.05$, 1000 simulations, Supplementary Fig. 10b), implying that the frequent disruption of CTCF by schizophrenia risk SNPs may be due to random effect.

There are several limitations in this study. First, we only used ChIP-Seq experiments from brain tissues (or neuronal cells) in

this study. Considering the fact that most ChIP-Seq experiments in ENCODE were performed in non-brain tissues, only limited TFs (i.e., 34) were included in this study. Therefore, risk SNPs that disrupt binding of other TFs could not be identified in this study. Second, potential target genes of 97 TF binding–disrupting SNPs were identified using eQTL annotation. Nevertheless, in most cases, the identified TF binding–disrupting SNP is associated with the expression of several genes. It remains largely unknown if all of the target genes or only a specific target gene have a role in schizophrenia. Third, considering that the reporter gene assays are relatively labor consuming and time costing, we only tested 19 identified functional SNPs (10 SNPs from functional annotation and 9 SNPs from functional genomics). Thus, only a small proportion of the 132 TF binding–disrupting SNPs were verified with reporter gene assays. Fourth, though we identified several potential causal risk variants in this study, we still do not know how this kind of genetic controls of gene expression confer schizophrenia risk. It is likely these identified risk variants confer schizophrenia risk through affecting gene expression (i.e., these identified SNPs affect the expression level of target genes, and the change of target gene expression may play a pivotal role in schizophrenia pathogenesis). However, more work is needed to further demonstrate how this kind of genetic controls of gene expression confer schizophrenia risk.

In summary, we generated the landscape of the plausible causal variants in schizophrenia for the first time and revealed the gene regulatory mechanisms affected by schizophrenia risk SNPs. Our study provides new insights into the genetic mechanisms of schizophrenia. Further mechanistic investigation and functional characterization of the identified causal variants and genes will help us understand the pathogenesis of schizophrenia.

## Methods

**Schizophrenia GWAS and potential causal SNPs.** Genetic associations from three large-scale GWASs[12–14] were used in this study. The first GWAS was from PGC. PGC performed a large-scale GWAS of schizophrenia (PGC2 release) and identified 128 genome-wide significant (GWS) regions (spanning 108 independent loci)[12]. For each of the GWS regions, an index SNP was defined (usually the most significant SNP) and a potential causal set of SNPs were identified. There is a 99% possibility that the causal variants were included in the defined potential causal set of SNPs. In addition to the 128 genome-wide significant regions, PGC2 also identified 141 genomic regions that showed suggestive associations ($P < 1.0 \times 10^{-6}$) with schizophrenia. Potential causal SNPs were also identified for these 141 suggestive regions. A total of 20,374 potential causal variants (including SNPs, insertions and deletions) from 125 GWS regions (three GWS regions on X chromosome were not included in this study, as potential causal SNPs were not defined by PGC2 for these three regions) and 141 suggestive risk regions were identified by PGC2. As we focused on SNPs, we excluded insertions and deletions. A total of 18,707 potential causal SNPs from PGC2 were retained for further analysis. More detailed information about the definition and identification of potential causal SNPs can be found in the original paper[12]. The second GWAS was from a recent study of Li et al.[13]. Li et al.[13] first conducted a GWAS in Chinese population (7699 cases and 18,327 controls). They then carried out a meta-analysis (43,175 cases and 65,166 controls) through combining the associations from Chinese population and PGC2. Finally, they performed replication study using an independent Chinese sample (4384 cases and 5770 controls). Through integrating the results from above three stages, they identified 30 novel schizophrenia risk loci. We extracted the SNPs that were in LD with the index SNPs ($r^2 > 0.3$) using genotype data (379 European individuals) from the 1000 Genomes Project (Phase I data, phase1_v3.20101123). PLINK[32] was used to calculate the LD values ($r^2$) between the index SNPs and nearby SNPs with following parameters: –ld-window-kb 1000, –ld-window 100000, –ld-window-r2 0.3. A total of 4794 SNPs (in LD with the 31 index SNPs) were identified. The third GWAS was from the study of Pardinas et al.[14] who identified 50 novel schizophrenia risk loci through using 40,675 cases and 64,643 controls. Pardinas et al.[14] used FINEMAP[26] to identify the potential causal SNPs and they identified 1799 potential causal SNPs for the 50 newly identified risk loci. As some of the potential causal SNPs from above three GWASs were overlapping, we removed the overlapping SNPs (Supplementary Fig. 11). A total of 23,400 non-overlapping SNPs were included in this study. To identify the most possible causal (or functional) SNPs at each of the GWAS risk loci, the potential causal SNPs from above three GWASs were subjected to deep bioinformatic and functional genomics analyses in this study.

**Prioritization of functional SNPs using different annotation approaches.** To prioritize the most possible functional SNPs at each of the risk loci, we used five well-characterized functional annotation tools, including CADD[27], Eigen[29], GWAVA[28], LINSIGHT[31] and RegulomeDB[30]. CADD uses evolutionary conservation information and data from ENCODE[44] to annotate and identify the possible causal or pathogenic variants. The scores of CADD range from 0 to 99. Eigen utilizes an unsupervised spectral approach to score the potential functional variants. The scores of Eigen range from −4.2 to 175.4. GWAVA annotates functional variants through using a wide range of data from genomic and epigenomic annotations, including open chromatin, TF binding, histone modification, conservation CpG islands and so on. The scores of GWAVA range from 0 to 1. LINSIGHT predicts the functional variants through combining functional genomic data with molecular evolution model. The scores of LINSIGHT range from 0 to 1. RegulomeDB annotates functional variants using a variety data from ENCODE, including ChIP-Seq, FAIRE, DNase I hypersensitive sites and eQTL data. The ratings of RegulomeDB range from 1 to 6. For CADD, Eigen, GWAVA and LINSIGHT, the larger the score, the higher probability that the SNP is functional. Therefore, the SNP with the highest score was defined as top functional SNP. For RegulomeDB, smaller rating suggests higher probability that the SNP is functional. Thus, the SNP with the smallest rating was defined as top functional SNP. As two prioritization strategies (CADD, Eigen, GWAVA and LINSIGHT use scores and RegulomeDB uses rating) were used by the functional annotation approaches, we defined the overlapping top functional SNPs as follows: (1) if a SNP has the highest score in at least two scoring annotation methods (CADD, Eigen, GWAVA and LINSIGHT), this SNP was defined as the top functional SNP; (2) assuming a SNP (e.g., SNP X) has the highest score in at least one scoring annotation method (CADD, Eigen, GWAVA and LINSIGHT), if this SNP also has the smallest rating in RegulomeDB, this SNP was defined as overlapping top functional SNP. Of note, as our goal is to identify the most possible functional SNP at each risk locus, we did not compare the scores of the top functional SNPs from different loci. More detailed information about these annotation approaches can be found in previous publications[27–31].

**Regulatory SNP annotation using functional genomics.** We downloaded ChIP-Seq data of 34 TFs from the ENCODE[44] (http://www.encodeproject.org). As schizophrenia is a mental disorder that mainly originates from dysfunction of brain, only ChIP-Seq experiments performed in human brain tissues or neuronal cells (including neuronal cell lines) were included. The detailed information about the TFs and ChIP-Seq experiments can be found in the Supplementary Table 4. We annotated regulatory SNPs as previously described[38]. Briefly, FastQC (http://www.bioinformatics.babraham.ac.uk/projects/fastqc) was used to check the quality of the raw and processed reads. Btrim[59] was used for quality control and filtering of the raw reads (with parameters "-a 20 -l 20"). We used cutadapt[60] to filter the over-represented sequences (including adaptors, primers and other sequences). The processed ChIP-Seq reads were then mapped to the human reference genome (GRCh37/hg19, which was used throughout all analyses) and bowtie[61] was used to perform alignment (with the parameters "-n 2 -e 70 -m 2 -k 2"). Peak calling was performed using MACS[62], with following parameters: "–keep-dup=1 -f BAM -w -S –call-subpeaks -g hs". For control experiments, if there were biological replicates, we chose the control experiment with the largest size of bam file. For ChIP-Seq experiments, if there were biological replicates, we combined the bam files and conducted peaks calling. The called peaks were then used for PWM identification.

To obtain DNA binding motifs (PWMs) enriched in the genomic sequences surrounding ChIP-Seq peaks, we utilized MEME[45] to run motif discovery on the sequences ±20 bp from the top 500 peaks (ranked by peak height), with parameters "-nmotifs 5 -minw 6 -maxw 20", and the background model was a 0-order Markov model. We compared the peaks with the corresponding control sample and filtered peaks with FDR > 5%. As a result, 30 TFs have peaks with FDR ≤ 5% (CHD2, IRF3, RCOR1 and USF2 were excluded due to limited or no peaks). To model the binding specificity of TFs, Whitington et al.[38] compiled a PWMs database that contains 7699 PWMs, including PWMs from the JASPAR, Uniprobe, Hi-SELEX and ChIP-Seq data. We compared the identified DNA binding motifs (PWMs) (derived from the ChIP-Seq experiments) with the PWMs in PWM database[38], and the matched PWMs (usually, the MOTIF1) (the most statistically significant, $E$-value of which is the smallest, motifs with small $E$-values are very unlikely to be random sequence artifacts) were used to annotate if the potential causal SNPs disrupted binding of TFs.

Potential causal SNPs that resided within ±50 bp of all ChIP-Seq peak summits (with a FDR ≤ 0.05) were identified and FIMO[46] was used to scan the occurrences of a given PWM within the genomic sequence overlapping a given SNP (with the parameter "–thresh 0.001"). We extracted the sequence ±20 bp from each potential causal SNP (both or more alleles) and scored every genomic position in which the SNP position overlapped the PWM by at least one base-pair. Both strands (with the reference allele and alternative allele) were considered. If one or more alleles of a SNP had a FIMO LLR (log-likelihood ratio) $P < 1 \times 10^{-3}$, the SNP was defined to disrupt the PWM. More information about FIMO can be found in the original paper[46] and FIMO website (http://mccb.umassmed.edu/meme/doc/fimo.html).

**DNase-Seq and histone modification data.** The DNase-Seq data were downloaded from the UCSC genome browser (http://hgdownload.cse.ucsc.edu/

goldenPath/hg19/encodeDCC/wgEncodeUwDnase/). Neuronal cells used for DNase-Seq were BE2C, HAc, SK-N-MC and SK-N-SH. Histone modification data were downloaded from ENCODE. Histone modifications, including H3K4me1, H3K4me3, H3K9ac, H3K9me3, H3K27ac, H3K27me3 and H3K36me3 were from the middle frontal area 46.

**Brain eQTL annotation**. To identify the potential target genes regulated by the identified regulatory SNPs, we examined three brain eQTL databases, including CMC[35], LIBD brain eQTL browser (http://eqtl.brainseq.org/phase1/eqtl/)[33] and GTEx[34]. To identify the genes that were differentially expressed in schizophrenia cases and controls, CMC collected brain tissues (dorsolateral prefrontal cortex (DLPFC)) of 258 schizophrenia cases and 279 controls[35]. Gene expression was determined using RNA sequencing and genotyping was performed with the Illumina Infinium HumanOmniExpressExome array. After stringent quality control, expression and genotype data from 467 subjects were used to perform eQTL analysis (utilizing an additive linear model implemented in MatrixEQTL). We used the eQTL data from these 467 individuals to explore if the regulatory SNPs were associated with gene expression in human brain. More detailed information about the sample collection, DNA and RNA extraction, gene expression quantification, genotyping, quality control and statistical analyses can be found in previous study[35]. The GTEx (Genotype-Tissue Expression) collected multiple tissues from healthy subjects. A total of 13 brain tissues (sample size ranges from 80 to 154) were included in the GTEx (Supplementary Table 3). Gene expression was quantified with RNA sequencing and genotyping was performed using Illumina OMNI 5M SNP Array. FastQTL was used to perform the eQTL analysis with following covariates, top 3 genotyping principal components, Genotyping array platform and sex. In addition, covariates identified by the probabilistic estimation of expression residuals (PEER) method were also included. We used brain eQTL data in this study. More information about the GTEx can be found in previous studies[34,50] and the GTEx website (https://gtexportal.org/home/). The LIBD eQTL browser (http://eqtl.brainseq.org/phase1/eqtl/)[33] contains brain expression data (from the DLPFC, quantified with RNA sequencing) and genotype data of 412 subjects (including 175 schizophrenia cases and 237 controls). Five levels of expression (including gene, exon, junction, transcript and expressed region) were quantified and only gene-level eQTLs were used in this study. The additive genetic effect model was used to test if a SNP was associated with the expression level of a gene, with adjusting for sex, ancestry, and expression heterogeneity.

**Distribution of random SNPs**. We first downloaded the PGC2 GWAS SNPs (a total of 8,624,491 SNPs with rs ID) from PGC2 website (https://www.med.unc.edu/pgc/results-and-downloads/). We then randomly sampled 186 SNPs from the PGC2 GWAS SNPs each time. We sampled 1000 times in total and SNPs sampled from these 1000 times were used for genomic location annotation. We used ANNOVAR[47] to annotate the genomic location of the sampled SNPs.

**Simulations**. We performed simulations to test if the 132 TF binding–disrupting SNPs were significantly associated with gene expression in human brain compared with random SNPs. Briefly, we randomly selected 132 SNPs (the number of SNPs equals to the identified TF binding–disrupting SNPs, and the minor allele frequencies of the random SNPs were matched to the 132 TF binding–disrupting SNPs) from PGC2 SNP list (9,444,230 SNPs in total) and calculated the number of the SNPs that showed significant association with gene expression in CMC eQTL results (FDR < 0.05). We conducted 10,000 simulations and calculated the mean and standard deviation of the 10,000 simulations. The significance level of our observation (i.e., 87 out of 132 SNPs were significantly associated with gene expression in CMC (FDR < 0.05)) were measured by $Z$ value as follow: $Z =$ (87–mean of the 10,000 simulations)/standard deviation of the 10,000 simulations. $Z$ value were converted into $P$ value with R command pnorm (-abs($Z$)).

**LD analysis between the 132 TF binding–disrupting SNPs**. LD analysis between the 132 TF binding–disrupting SNPs was performed using SNiPA[49]. We used default settings of SNiPA (genome assembly:GRCh37; variant set:1000 Genome, Phase 3 v5; population: Europeans) to calculate the LD values between the 132 TF binding–disrupting SNPs. More detailed information about SNiPA can be found in the original paper[49] and the SNiPA website (http://snipa.helmholtz-muenchen.de/snipa3/).

**Validation of the eQTL results with allele-specific expression analysis**. We verified the eQTL associations using ASE data from the GTEx (phs000424.v7.p2)[50]. The GTEx contains genotypes data and expression data (quantified with RNA sequencing) from 714 subjects and 53 tissues. As we focused on brain tissues, we only extracted the ASE results from brain tissues. Binomial tests were used to determine if the ratio of the two alleles was significantly different from the expectation[50] and regulatory SNPs with a binomial $P < 0.001$ were extracted (regulatory SNPs that showed allele-specific expression (or allelic imbalance) were defined at binomial $P < 0.001$ threshold). More information on ASE analysis can be found in the original paper[50] and GTEx website (https://gtexportal.org/home/).

**Cell culture**. All of the cell lines used in this study were purchased from the Kunming Cell Bank, Chinese Academy of Sciences (these cell lines were originally obtained from the ATCC). HEK293 cells were cultured in high-glucose Dulbecco's Modified Eagle's medium (DMEM; C11995500BT, Gibco) containing 10% fetal bovine serum (FBS; 10091148, Gibco), penicillin and streptomycin (100 U/ml)) (10378016, Gibco). SK-N-SH and SH-SY5Y cells were cultured in high-glucose DMEM (C12430500BT, Gibco) supplemented with 10% FBS, 10 mM sodium pyruvate solution (11360070, Gibco), 1× Minimum Essential Medium non-essential amino acid solution (11140050, Gibco), penicillin and streptomycin (100 U/ml). The antibiotics were withdrawn 48 h before performing assays and all cells were cultured at 37 °C with 5% $CO_2$ and 95% air. No mycoplasma contamination was found for the cell lines used in this study.

**Vector construction**. The DNA sequence (length ranges from ~300 to 700 bp) containing the test SNP was amplified using primers (Supplementary Table 10) linked with homologous arms (which were identical with the sequence (located at the multiple clone sites) of pGL3-promoter vectors) (Supplementary Fig. 12) and PCR products were purified with DNA Purification Kit (DP209, TIANGEN). We digested pGL3-Promoter vector (E1761, Promega) and pGL4.11 vector (E6661, Promega) with *Kpn*I (FD0524, FastDigest) and *Xho*I (FD0694, FastDigest), and the digestion products were purified with DNA Purification Kit. Then the purified DNA fragments containing the test SNP were inserted into the pGL3-Promoter vector using the Trelief™ SoSoo Cloning Kit (TSV-S1, TSINGKE). As SNP rs2270363 is located in promoter region, we inserted the DNA fragments containing rs2270363 into the pGL4.11, which is a basic vector with no promoter. The ligated vectors were then used to transform DH5α competent cells (Takara). LB plates (with the selectable marker, ampicillin) were used to select the transformed cells and recombinant plasmids were extracted from the transformed cells grown from single colony. PCR-mediated point mutation technique was used to generate the DNA fragments containing the alternative allele of this test SNP. All sequences of the inserted DNA fragments were verified using Sanger sequencing.

**Cell transfection and dual-luciferase reporter gene assays**. The constructed vector (contained the test SNP) (150 ng for 96-well plate and 500 ng for 24-well plate) and internal control plasmid pRL-TK (E2241, Promega) (30 ng for 96-well plate and 50 ng for 24-well plate) were co-transfected into the tested cell lines. HEK293 cells were transfected with the PEI method[63], and SK-N-SH and SH-SY5Y cells were transfected using the Lipofectamine 3000 (L3000-015, Invitrogen), according to the manufacturer's instructions. For SK-N-SH and SH-SY5Y cells, dual-Luciferase reporter gene assays were performed in a 96-well white plate containing 150 µl medium at $2 \times 10^5$ cells/ml. For HEK293 cells, reporter gene assays were performed in a 24-well white plate containing 500 µl medium at $5 \times 10^5$ cells/ml. After 48 h post transfection, Luminoskan Ascent instrument (Thermo Scientific) was used to measure the luciferase activity with the Dual-Luciferase Reporter Assay System (E1960, Promega). All of the experiments were performed according to the instructions recommended by the manufacturer. The luciferase activity data were obtained from at least eight replicate wells. Two-tailed Student's $t$-test was used to compare if the difference was significant and the significance threshold was set at $P < 0.05$.

**Gene Ontology analysis**. GO analysis was performed using DAVID (v6.8)[51]. Three GO terms (including biological process (GOTERM_BP_DIRECT), molecular function (GOTERM_MF_DIRECT) and cellular component (GOTERM_CC_DIRECT)) were used to test if specific categories or pathways were enriched among the target genes of the identified regulatory SNPs. The significance ($P$ value) of the overrepresented GO terms was corrected by the Benjamini–Hochberg procedure.

**Spatio-temporal expression pattern analysis**. Spatio-temporal expression pattern analysis was conducted as previously described[52,64]. Briefly, expression data from different developmental stages of human brain were downloaded from the BrainSpan (http://www.brainspan.org/). The expression level of each gene was quantified (expressed as RPKM (read per kilobase per million)) with RNA sequencing. For a specific development stage, the median expression level of all genes in a gene set (target genes was treat as a gene set in this study) was used to represent the expression level of the gene set at this stage. Three gene sets (genes in Supplementary Table 5, Supplementary Data 4, and background genes) were used in this study. Background genes were extracted from a previous study[65]. Wilcoxon rank sum test (implemented in R software (v3.5.0)) was used to compare if the expression level of target genes and background genes was statistically significantly different.

**Cell type-specific expression analysis of target genes**. Cell type-specific expression data were extracted from the study of Skene et al.[53]. Skene et al.[53] performed cell type-specific expression analysis using single cell sequencing from mouse brain (a total of 9790 cells from different brain regions, including neocortex, striatum, midbrain, hippocampus, etc) and they calculated the specificity (higher specificity of a gene suggests the expression of this gene is enriched in a specific cell type, i.e., this gene is specifically expressed) value of each gene in each cell type. We extracted the specificity value of the target genes of the identified regulatory SNPs

and counted the genes with a specificity value >0.10. Detailed information about the single cell sequencing and calculation of the specificity can be found in the study of Skene et al.[53].

**Comparison of the number of binding sites of TFs.** Frequent disruption of CTCF and POLR2A binding (by schizophrenia risk SNPs) was observed in our study. A possible explanation for this observation is that CTCF and POLR2A have more binding sites on the genome than other TFs. Thus, the number of SNPs that disrupt CTCF and POLR2A binding were accordingly larger compared to other TFs. To test if POLR2A and CTCF have more binding sites on the genome, we performed additional analyses. We first identified ChIP-Seq peaks of each TF (as described in previous section). We identified 8447 SNPs that disrupted binding of 30 TFs (these 30 TFs were used in this study) through mapping a total of 968,903 SNPs (from the Illumina HumanOmni1-Quad, only SNPs with rs id were retained) to the identified ChIP-Seq peaks (the identification of TF binding–disrupting SNPs was described in more detail above). We then randomly sampled 132 SNPs (the number of SNPs equals to the identified TF binding–disrupting SNPs in this study) from the 8447 TF binding–disrupting SNPs and counted the number of SNPs that disrupted the binding of each TF. We conducted 1000 simulations and calculated the mean and standard deviation of the 1000 simulations. We plotted the distribution of the number of TF binding–disrupting SNP using ggplot2 package (implemented in R Software). In addition, we also compared the number of motifs (appear on the genome) of the 30 TFs through counting the identified ChIP-Seq peaks of each TF.

**URLs.** For SZDB, see http://www.szdb.org/; for PGC2, see http://www.med.unc.edu/pgc/; for CADD, see http://cadd.gs.washington.edu/; for GWAVA, see https://www.sanger.ac.uk/sanger/StatGen_Gwava; for Eigen, see http://www.columbia.edu/~ii2135/eigen.html; for LINSIGHT, see https://github.com/CshlSiepelLab/LINSIGHT; for RegulomeDB, see http://www.regulomedb.org/; for CMC, see http://www.synapse.org/CMC; for GTEx, see https://gtexportal.org/home/; for LIBD brain eQTL browser, see http://eqtl.brainseq.org/phase1/eqtl/; for ENCODE, see https://www.encodeproject.org/; for BrainSpan, see http://www.brainspan.org/; for MEME, see http://meme-suite.org/tools/meme; for FIMO, see http://meme-suite.org/tools/fimo; for MACS, see http://liulab.dfci.harvard.edu/MACS/; for Bowtie, see http://bowtie-bio.sourceforge.net/index.shtml; for FastQC, see http://www.bioinformatics.babraham.ac.uk/projects/fastqc/; for Cutadapt, see https://cutadapt.readthedocs.io/en/stable/index.html; for The 1000 Genomes Project, see http://www.1000genomes.org/; for PLINK, see http://zzz.bwh.harvard.edu/plink/; for SNiPA, see http://snipa.helmholtz-muenchen.de/snipa3/.

## Data availability

The PWM, ChIP-Seq, DNase-Seq and histone modification data of the 132 TF binding–disrupting SNPs can be accessed and visualized at SZDB database[25] (http://www.szdb.org/). The source data underlying Figs. 2b, 4c–e, 5b–d, f–h, 6b–d, f–h, 7b, c, e, f and 8a–j and Supplementary Figs. 4b–d and 5b–d are provided as a Source Data file. The rest of the data (including custom code and plasmids) are available from the corresponding author upon request.

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

## Acknowledgements

This study was equally supported by the Strategic Priority Research Program of the Chinese Academy of Sciences (XDB13000000) and the National Key Research and Development Program of China (Stem Cell and Translational Research) (2016YFA0100900) (to X.-J.L.). It was also supported by the National Nature Science Foundation of China (31722029 to X.-J.L.) and the Key Research Project of Yunnan Province (2017FA008 to X.-J.L.). We thank Dr. Whitington for sharing the curated PWM data with us. One of the brain eQTL datasets used in this study were generated as part of the CommonMind Consortium supported by funding from Takeda Pharmaceuticals Company Limited, F. Hoffman-La Roche Ltd and NIH grants R01MH085542, R01MH093725, P50MH066392, P50MH080405, R01MH097276, RO1-MH-075916, P50M096891, P50MH084053S1, R37MH057881 and R37MH057881S1, HHSN271201300031C, AG02219, AG05138 and MH06692. Brain tissue for the study was obtained from the following brain bank collections: the Mount Sinai NIH Brain and Tissue Repository, the University of Pennsylvania Alzheimer's Disease Core Center, the University of Pittsburgh NeuroBioBank and Brain and Tissue Repositories and the NIMH Human Brain Collection Core. CMC Leadership: Pamela Sklar, Joseph Buxbaum (Icahn School of Medicine at Mount Sinai), Bernie Devlin, David Lewis (University of Pittsburgh), Raquel Gur, Chang-Gyu Hahn (University of Pennsylvania), Keisuke Hirai, Hiroyoshi Toyoshiba (Takeda Pharmaceuticals Company Limited), Enrico Domenici, Laurent Essioux (F. Hoffman-La Roche Ltd), Lara Mangravite, Mette Peters (Sage Bionetworks), Thomas Lehner, Barbara Lipska (NIMH). The Genotype-Tissue Expression (GTEx) Project was supported by the Common Fund of the Office of the Director of the National Institutes of Health, and by NCI, NHGRI, NHLBI, NIDA, NIMH and NINDS.

## Author contributions

X.-J.L. conceived and designed the study. Y.H. performed most of the bioinformatic analyses, including the processing of the raw ChIP-Seq data, the identification of PWMs from the ChIP-Seq peaks and the identifications of TF binding–disrupting SNPs. S.L. performed the reporter gene assays. X.L. carried out the functional SNP annotations (using CADD, GWAVA, Eigen, LINSIGHT and RegulomeDB) and eQTL annotation. J.L. conducted spatio-temporal expression pattern, cell type-specific expression analysis and simulations. Y.H., S.L., J.L., X.L. and X.-J.L. contributed to this work in data generation and analysis, result interpretation and manuscript writing. X.-J.L. oversaw the project and drafted the first version of the manuscript. All authors revised the manuscript critically and approved the final version.

## Additional information

**Competing interests:** The authors declare no competing interests.

