## [Peer Review File · Nature Communications]

Reviewer #1 (Remarks to the Author):

In the study, Huo et al. combined analytical approaches with functional genomics to identify the causative genetic variants underpinning schizophrenia risk. The methods are sound, the analyses were carefully conducted and the paper is well written. The methodology developed in this study can be applied in principle to all complex traits and disease conditions. However, I have several concerns about the structure of the paper and some of the analyses.

The first part of the paper that uses functional annotation tools to annotate the credible schizophrenia SNPs followed by reporter gene assays is slightly disconnected with the rest of the paper.

The authors need to demonstrate whether the allelic effects on gene expression observed in reporter gene assays are consistent with the estimated eQTL effects of the corresponding alleles in GTEx or any other brain eQTL data sets.

Although the authors have used reporter gene assays to verify the causal variants for some of the genes, it remains unclear how this kind of genetic controls of gene expression confer schizophrenia risk. This is probably one of main limitations of this study.

Page 22 "... 58 showed significant association with the expression of the same gene in at least two independent eQTL datasets (Table 1), suggesting these genes are true targets of the identified regulatory SNPs ..."

Page 23 "... genes in independent brain eQTL datasets strongly suggest that these genes are true targets for the identified regulatory SNPs."

It is possible that those regulatory SNPs are just in LD with the actual causal variants unless they have been confirmed by the reporter gene assays. I would not be surprised if a "regulatory SNP" is associated with multiple genes in the region and some of the associations could be even stronger than that with the target gene.

It might be useful to show the associations of the regulatory SNPs with all genes in close physical proximity e.g. +/- 2Mb.

Page 26 "In addition, we also found that the expression level of the target genes was higher in prenatal stage compared with postnatal stage ($P=0.001$ for gene set in Table 1 and $P=0.008$ for gene set in Table S8) (Figure 8k), suggesting that the target genes may play a role in brain development."

It is not clear how this analysis was performed. Note that gene expression in different developmental stages are correlated so that the analysis to test for the difference between prenatal and postnatal stages needs to take the covariance into account.

Minor concerns

Page 6 "... and found that 153 loci contained overlapping top functional SNPs ...".

What is the definition of a locus and what is the average number of SNPs at each of the loci? The total number of SNPs were filtered down from ~24K to 153. Have the authors thought about whether the filtering criteria were too stringent?

Page 6 "... prioritized by at least two independent annotation methods ..."

This is slightly arbitrary because the methods used in some of the tools are similar so that some

degree of overlap may be expected even for random SNPs.

Page 6 “Most of the prioritized functional SNPs were located in intergenic and intronic regions (Fig. 2a).”

What is the distribution of random SNPs in the annotated regions as shown in Fig. 2a?

Page 8 “... 66 prioritized top functional SNPs showed significant association ...”

Significant after correcting for multiple testing?

Figure 2b.

Strictly speaking, only 5 of the SNPs are significant after correcting for multiple tests (i.e., $p < 0.05 / 10$).

Figure 3.

Any comments why ~50% of the SNPs are intronic?

Figure S1.

The GWAS signal is only marginally significant. Is it genome-wide significant in the other two GWAS data sets?

Page 17 “... USF1 and MAX1 binding sites. We validated the regulatory effect of rs2270363 with reporter gene assays”

Please clarify which gene(s) were tested in the reporter gene assays?

Page 21 “... enhanced the activity of luciferase compared with control vector (empty pGL3-promoter) in the tested cell lines (Fig. 4-7, Supplementary Fig. 2 and 3), supporting that these SNPs were located in enhancer regions.”

The binding proteins can be repressors (e.g. ARID5B downregulates IRX3 and IRX5; Claussnitzer et al. 2015 NEJM).

Page 22 “... brain (FDR<0.05 in CMC, FDR<0.01 in LIBD, and P<0.001 in GTEx) ...”

It is not justified to use different p-value thresholds in different data sets.

Page 23 “Compared with the random SNPs, the identified regulatory SNPs showed significant association with gene expression ($Z=18.5$, $P=1.44E-76$), ...”

It might be useful to show a figure here.

Figure 8 “... ($P<1.0\times 10^{-7}$ for gene set in Table 1 and $P<5.0\times 10^{-7}$ for gene set in Supplementary Table 8).”

The difference between Table 1 and Table S8 need to be clarified.

Page 29 “. CTCF is a transcriptional repressor encoded by CTCF gene, which has pivotal roles in transcription regulation. ”

Is the expression level of CTCF associated with any SNPs in cis? If so, those SNPs are expected to be associated with SCZ.

Page 31 “... using genotype data (379 individuals) from the 1000 Genomes Project (Phase I data, phase1_v3.20101123).”

Chinese, Europeans or all the 1000 Genomes samples?

Page 32 “As some of the credible causal SNPs from above three GWASs were overlapping, we removed the overlapping SNPs. A total of 23,400 non-overlapping ... ”

A Venn diagram to show the overlaps?

Page 36 “... selected 132 SNPs (the number of SNPs equals to the identified TF binding-disrupting SNPs) from PGC2 SNP list (9,444,230 SNPs in total) ...”

With minor allele frequencies of the SNPs matched to the 132 TF binding-disrupting SNPs?

Reviewer #2 (Remarks to the Author):

The manuscript by Huo et al provides comprehensive, mostly bioinformatic analyses of a large number schizophrenia associated GWAS loci aiming to pinpoint the causal variant(s). The strength of the study is that for a few loci they have extended the bioinformatic analyses to wet lab validation experiments. The work has been performed quite comprehensively.

1) The main drawback with the current version of the manuscript is that it is not easy to extract what they actually found. Table 1 tries to summarize this, but is so comprehensive that it does not highlight the main findings. This should be clarified.

2) It is also hard to follow the number of associated SNPs, for example:

a. On line 26 and 27 in the abstract (and line 347) they state that “We found 97 of the 132 TF binding-disrupting SNPs are associated with gene expression...”

b. On lines 134 and 135 they state that “In total, 66 prioritized top functional SNPs showed significant association with gene expression...”. This is the only place in the manuscript where the number of 66 associated SNPs is mentioned

Even after reading the manuscript several times it is hard to follow the numbers. I suggest that the authors prepare a figure/table that summarizes to the reader the main results to complement the current Fig 1 that demonstrates the flow of the analyses. Fig 1 clarifies nicely the flow of analyses, but it only partially clarifies how many SNPs at each point had a positive finding in expression analyses.

3) The authors highlight the reporter gene assay studies, which are a good extension. However, understandably only 10 of the 66 SNPs (as stated on line 137, yet line 330 indicates that 9 SNPs were tested, please clarify) were tested in reporter assays. The abstract is misleading, it states on line 27 that “we validated the regulatory effect of the identified SNPs with reporter gene assays and..... “, this is an overstatement as only 10 (or 9, whichever is correct) was tested. The fact that only a small fraction of loci were validated should be stated more clearly throughout the manuscript.

4) The fourth main feature of the manuscript is that they only analyzed brain tissue. It is not surprising that they then find variants associated to neural tissue. This was a deliberate choice, which should be addressed more clearly in the text. In other words, it remains unclear how specific these findings are to neural tissues, compared to other tissues.

5) It would be good to add a list of limitations of the study. Now it reads as slightly too comprehensive and does not give a reality check.

Reviewer #3 (Remarks to the Author):

In this manuscript, the authors aim to identify SNPs that have causal functions in relation to SCZ based on GWAS findings. The authors first prioritize SNPs based on either well-established annotations (CADD, Eigen, GWAVA, RegulomeDB and LINSIGHT) or TFBS using Chip-Seq data. They then test several prioritized SNPs by reporter gene assay to further support their findings. The authors claim that frequent disruption of CTCF is suggestive of an important role in schizophrenia.

The authors used a bioinformatics approach to integrate publicly available data sets with functional experiments to support their findings. However, the study seems largely based on arbitrary decisions without proper justification. In addition, this type of analysis, especially for SCZ, has been widely performed but the authors barely compare their work with previous findings. The reported findings from pathway/gene expression analyses are not novel.

Below are my main comments followed by some minor comments.

1. Authors should clarify why they conducted two analyses with well-established annotations and Chip-seq, separately. This also makes the manuscript a little bit hard to follow. In addition, I believe some of the annotations such as CADD, RegulomeDB and GWAVA include Chip-seq data. I would expect to find overlap of prioritized SNPs between these two analyses but the authors did not discuss that. I would suggest to integrate both annotations as they can support each other. Otherwise the authors should clearly state why it is better to keep separate.
2. In the section of "Prioritization of functional SNPs and experimental verification", the authors compared top functional SNPs per locus but it is not clear what are the "top functional SNPs". It should be clarified, in the method section, what the scale of the scores is and the threshold for each annotation. One concern is that, the score of "top SNPs" can vary highly per locus. For example, when the top score in locus A is 100, the top score in locus B can be 10 which is much less likely to be functional compared to locus A. The authors need to clarify how they dealt with this.
3. Analyses related to fig. 2b: it is not clear how the authors selected the 10 SNPs
4. For 132 SNPs disrupting TFBSs, it is crucial to check their LD dependence. The authors should report how many of them are independent from 132. For example, they found 6 SNPs that had significant association with FAM109B but those SNPs can be in LD and those associations might thus not be independent.
5. Authors mentioned that POLR2A and CTFT have the largest number of SNPs but that may be because those TFs regulates more genes than other TFs. Authors should show if those findings are not random.
6. Reporter gene assay was performed only for several SNPs from 132 SNPs. The authors should justify how they selected those SNPs and why they only tested these.
7. In the paper, the authors performed reported gene assay to assess the regulatory function of prioritized SNPs but target genes are identified based on eQTLs annotations. eQTLs have been used previously to identify target genes from SCZ risk loci (e.g. Gusev, A. et al. Nat Genet. 2018), indeed findings from pathway enrichment and gene expression are already known (e.g. Pers, T. Human Mol. Genet. 2016, Won, H. et al. Nature. 2016). Authors should discuss what is novel and what is known already.

Minor comments

1. The word “credible causal SNPs” is often used to refer to SNPs after in silico fine mapping. There is no official rule about this but I would advise to use slightly different wording for the SNPs that are in LD with one of the GWAS hits, such as potential causal SNPs or candidate SNPs to avoid confusion.
2. The section “Validation of the regulatory effect of the identified TF binding-disrupting SNPs” does not seem to contain any new information but is just a summary of the previous sections. I am not sure why this section is necessary.
3. On line 345, “However” does not seem to be a correct conjunction in that sentence.
4. Authors mentioned “eQTL analysis” several times, but I assume publicly available summary statistics were used rather than performing actual eQTL analysis. To avoid confusion it should be rephrased to “eQTL annotation”.

We thank the reviewers greatly for their thorough review and highly appreciate the constructive comments and suggestions, which significantly contributed to improving the quality of this manuscript. Please find below a detailed response to each of the comments.

Reviewers' comments:

Reviewer #1 (Remarks to the Author):

In the study, Huo et al. combined analytical approaches with functional genomics to identify the causative genetic variants underpinning schizophrenia risk. The methods are sound, the analyses were carefully conducted and the paper is well written. The methodology developed in this study can be applied in principle to all complex traits and disease conditions.

Response: We thank the reviewer for the positive comment on our work.

However, I have several concerns about the structure of the paper and some of the analyses.

The first part of the paper that uses functional annotation tools to annotate the credible schizophrenia SNPs followed by reporter gene assays is slightly disconnected with the rest of the paper.

Response: We thank the reviewer for pointing this out. Following the reviewer's comment, we have revised the manuscript accordingly. The first part of the paper is logically closely connected with the rest of the paper in the revised manuscript. **Page 8, lines 7-22, Page 9, line 1.**

In addition, we also discussed why we performed functional annotation and functional genomics separately in the revised manuscript. **Page 23, lines 8-22, Page 24, lines 1-8.**

The authors need to demonstrate whether the allelic effects on gene expression observed in reporter gene assays are consistent with the estimated eQTL effects of the corresponding alleles in GTEx or any other brain eQTL data sets.

Response: We thank the reviewer for this valuable suggestion. According to the reviewer's suggestion, we compared the allelic effects on gene expression observed in reporter gene assays with the estimated eQTL effects of the corresponding alleles in CMC dataset. Among the 10 SNPs tested in Fig.2b, 7 SNPs (rs11655813, rs9908888, rs17821573, rs301791, rs37718, rs7304782, rs7752421) showed significant allelic effects in reporter gene assays. Among these 7 significant SNPs, 5 SNPs (rs11655813, rs9908888, rs17821573, rs301791 and rs7752421) were significantly associated (FDR<0.01) with gene expression in CMC dataset. For these 5 significant eQTL SNPs, we found that the allelic effects on gene expression observed in

reporter gene assays are consistent with the estimated eQTL effects of the corresponding alleles in CMC dataset. We have included this new result as a supplementary figure (**Supplementary Fig.1**) in the revised manuscript. **Page 7, lines 5-22.**

Although the authors have used reporter gene assays to verify the causal variants for some of the genes, it remains unclear how this kind of genetic controls of gene expression confer schizophrenia risk. This is probably one of main limitations of this study.

Response: We agree with the reviewer that we still do not know how this kind of genetic controls of gene expression confer schizophrenia risk. We identified several potential causal risk variants in this study and we speculated that these identified risk variants may confer schizophrenia risk through affecting gene expression (i.e., these identified SNPs affect the expression level of target genes, and the change of target gene expression may play a pivotal role in schizophrenia pathogenesis). However, more work is needed to further demonstrate how this kind of genetic controls of gene expression confer schizophrenia risk. We discussed these limitations in our revised manuscript. **Page 27, lines 1-18.**

Page 22 "... 58 showed significant association with the expression of the same gene in at least two independent eQTL datasets (Table 1), suggesting these genes are true targets of the identified regulatory SNPs ..."

Page 23 "... genes in independent brain eQTL datasets strongly suggest that these genes are true targets for the identified regulatory SNPs."

It is possible that those regulatory SNPs are just in LD with the actual causal variants unless they have been confirmed by the reporter gene assays. I would not be surprised if a "regulatory SNP" is associated with multiple genes in the region and some of the associations could be even stronger than that with the target gene.

Response: We agree with the reviewer that it is possible that those regulatory SNPs are just in LD with the actual causal variants unless they have been confirmed by the reporter gene assays. According to the reviewer's suggestion, we have revised these statements in the revised manuscript. **Page 15, lines 21-22, Page 16, lines 1-3.**

It might be useful to show the associations of the regulatory SNPs with all genes in close physical proximity e.g. +/- 2Mb.

Response: We thank the reviewer for this valuable suggestion. Following the reviewer's suggestion, we performed eQTL analysis (using CMC dataset) to show the associations of the regulatory SNPs with all genes in close physical proximity (e.g. +/- 2Mb). For each regulatory SNP (the TF binding-disrupting SNPs were listed in Table S7, a total of 132 SNPs), we performed eQTL through using the genotype and expression data from the CMC dataset. The eQTL analysis was conducted with PLINK (v1.9). The associations of the regulatory SNPs with all genes in close physical proximity (e.g. +/- 2Mb) were listed in Supplementary Table xxx (Of note, only associations with a $P < 0.01$ were showed). We have

included this new result as a supplementary Table (**Supplementary Table 9**) in the revised manuscript. **Page 16, lines 19-22, Page 17, lines 1-4.**

Page 26 “In addition, we also found that the expression level of the target genes was higher in prenatal stage compared with postnatal stage ($P=0.001$ for gene set in Table 1 and $P=0.008$ for gene set in Table S8) (Figure 8k), suggesting that the target genes may play a role in brain development.”

It is not clear how this analysis was performed.

Response: We are sorry for the confusion. To compare the expression level of the target genes in prenatal stages and postnatal stages, we used a method developed by Gilman *et al* (Gilman *et al.* 2012). Briefly, for a specific development stage, the median expression level of all genes in a gene set (target genes was treat as a gene set in this study) was used to represent the expression level of the gene set at this stage. Wilcoxon rank-sum test was used to compare the expression level of target genes in prenatal stages and postnatal stages. **Page 39, lines 4-13.**

Note that gene expression in different developmental stages are correlated so that the analysis to test for the difference between prenatal and postnatal stages needs to take the covariance into account.

Response: We understand the reviewer’s concern. However, as the covariates were not available, we carried out the gene expression analysis in different stages as previously described (Gilman *et al.* 2012).

Minor concerns

Page 6 “... and found that 153 loci contained overlapping top functional SNPs ...”.

What is the definition of a locus and what is the average number of SNPs at each of the loci?

Response: Regular LD clumping (implemented in PLINK (Purcell *et al.* 2007)) was performed to obtain the independent index SNPs (with following parameters: $r^2 < 0.1$, $P1 < 5 \times 10^{-8}$, window size < 500 kb). This procedure uses a greedy algorithm (index SNPs are chosen greedily, starting with the SNP that has the lowest P-value) to define index SNPs. First, identifying the SNP with the lowest P value (this SNP was referred to as the first index SNP). Second, clumping the SNPs that were in LD ($r^2 > 0.1$, $P1 < 5 \times 10^{-8}$, window size < 500 kb) with the index SNP. SNPs met the clumping criteria (i.e., $r^2 > 0.1$ with the index SNP, $P1 < 5 \times 10^{-8}$, window size < 500 kb) were clumped. Third, performing the LD clumping again (of note, the index SNP and nearby SNPs that were in LD with the index SNP ($r^2 > 0.1$ with the index SNP, $P1 < 5 \times 10^{-8}$, window size < 500 kb) were clumped in the first round. Thus, these SNPs were excluded from the second round of clumping) and identifying the second index SNP (the SNP with the lowest P value in the remaining SNPs). This greedy algorithm was repeated and all of the index SNPs were identified (until there are no SNPs

with P value less than 5×10^{-8}). Each SNP only appears once in a clump. Through using this approach, PGC2 identified 128 index SNPs from the associated regions.

PGC2 defined a locus with following criterial: the physical region containing all SNPs that were in LD (i.e., $r^2 > 0.6$) with each of the 128 index SNPs. Associated loci within 250 kb of each other were merged. Finally, 108 physically distinct associated loci were identified by PGC2.

The average number of SNPs at each of the loci is about 80. **Page 6, lines 11-19.**

The total number of SNPs were filtered down from ~24K to 153. Have the authors thought about whether the filtering criteria were too stringent?

Response: We understand the reviewer's concern. We used two methods to identify the potential causal SNPs at the reported risk loci and to reveal the gene regulatory mechanisms underlying schizophrenia. We first prioritized the most possible functional SNPs using functional annotation (including CADD, GWAVA, Eigen, LINSIGHT and RegulomeDB). A total of 153 top functional SNPs were prioritized by functional annotation. We then identified 132 SNPs that disrupt binding of transcription factors (TFs) using functional genomics. We further identify the potential target genes of the TF binding-disrupting SNPs. Our goal is not only to identify the most possible causal risk variants for schizophrenia, but also provide a candidate list (i.e., top functional SNPs) for further functional investigation. If we used a less stringent filtering criteria, the number of the prioritized SNPs will increase. Accordingly, this will increase the difficulty for SNP selection if we or other researchers want to validate the regulatory effects of the identified regulatory SNPs with functional experiments. Thus, we used a stringent filtering criteria.

Page 6 "... prioritized by at least two independent annotation methods ..."

This is slightly arbitrary because the methods used in some of the tools are similar so that some degree of overlap may be expected even for random SNPs.

Response: We thank the reviewer for pointing this out and we agree with the reviewer that some of the tools are similar. We have revised this statement accordingly in the revised manuscript. **Page 6, lines 11-19.**

Page 6 "Most of the prioritized functional SNPs were located in intergenic and intronic regions (Fig. 2a)."

What is the distribution of random SNPs in the annotated regions as shown in Fig. 2a?

Response: Following the reviewer's suggestion, we performed additional analysis. First, we downloaded the PGC2 GWAS SNPs (a total of 8,624,491 SNPs with rs ID) from PGC2 website (<https://www.med.unc.edu/pgc/results-and-downloads/>). We then randomly sampled 186 SNPs from the PGC2 GWAS SNPs each time. We sampled 1,000 times in total and SNPs sampled from these 1,000 times were used for genomic location annotation. We used

ANNOVAR (Wang et al. 2010) to annotate the genomic location of the sampled SNPs. The distribution of random SNPs is shown below:

Distribution of random SNPs in genomic regions

Fig. Most of the random SNPs are located in intergenic and intronic regions. Of note, more than 50% random SNPs are located in intergenic regions, which is slightly different from the distributions in Fig. 2a (most prioritized functional SNPs in Fig.2a were located in intronic regions). Page 10, lines 16-22, Page 11, lines 1-2, Page 35, lines 5-10.

Page 8 “... 66 prioritized top functional SNPs showed significant association ...”
Significant after correcting for multiple testing?

Response: To explore if the prioritized top functional SNPs are associated with gene expression, we examined three brain eQTL data sets (including CMC, LIBD brain eQTL browser and GTEx). We used the default significance level used in each eQTL data set (FDR<0.05 in CMC, FDR<0.01 in LIBD and P<0.01 in GTEx), so some of the associations will not survive for multiple testing. However, some of the associations are still significant after correcting for multiple testing. To list all of the possible targets genes for these prioritized top functional SNPs, we did not correct these associations for multiple testing. To make our description more clearly, we have revised this statement in the revised manuscript. Page 7, lines 5-8.

Figure 2b.

Strictly speaking, only 5 of the SNPs are significant after correcting for multiple tests (i.e., $p < 0.05 / 10$).

Response: We agree with the reviewer that only 6 SNPs (including rs11655813, rs9908888, rs17821573, rs301791, rs37718, and rs7752421) are significant after correcting for multiple testing. To make our description more accurate, we have amended this in the revised manuscript. **Page 7, lines 13-17.**

Figure 3.

Any comments why ~50% of the SNPs are intronic?

Response: We thank the reviewer for this question. We speculated that the potential reasons are as follows: First, human genome is mainly made up of non-coding sequences (including intronic and intergenic regions) (protein-coding sequences account for only a very small fraction of the genome (approximately 1.5-2.0%)), so most of the TF binding-disrupting SNPs we observed were located in intronic and intergenic regions (That is, our observation that ~50% of the TF binding-disrupting SNPs are intronic is only due to that intronic region is a major component of human genome). Second, this result suggests that intronic region may have pivotal roles in regulating gene expression. Third, this observation may also implies that genetic variants in intronic region may have important role in schizophrenia susceptibility.

Of note, we also selected random SNPs and generated the distribution of the random SNPs (please refer to previous comment). We found that most of the random SNPs were located in intergenic regions (but not intronic regions observed in our study), which suggests that causal risk variants for schizophrenia are enriched in intronic regions. However, more work is needed to validate our speculations. **Page 10, lines 16-22, Page 11, lines 1-2, Page 35, lines 5-10.**

Figure S1.

The GWAS signal is only marginally significant. Is it genome-wide significant in the other two GWAS data sets?

Response: We checked the association between SNP rs1801311 and schizophrenia in other two datasets. The *P* value of rs1801311 in study of Li et al. (Li et al. 2017) is 1.18×10^{-7} . However, in study of Pardinas et al. (Pardinas et al. 2018), the *P* value of rs1801311 is 1.47×10^{-9} , which exceeds genome-wide significance level (5.0×10^{-8}).

Page 17 "... USF1 and MAX1 binding sites. We validated the regulatory effect of rs2270363 with reporter gene assays"

Please clarify which gene(s) were tested in the reporter gene assays?

Response: We thank the reviewer for pointing this out. Following the reviewer's suggestion, we have revised this accordingly in the revised manuscript. **Page 13, lines 13-16.**

Page 21 “... enhanced the activity of luciferase compared with control vector (empty pGL3-promoter) in the tested cell lines (Fig. 4-7, Supplementary Fig. 2 and 3), supporting that these SNPs were located in enhancer regions. ”

The binding proteins can be repressors (e.g. ARID5B downregulates IRX3 and IRX5; Claussnitzer et al. 2015 NEJM).

Response: We thank the reviewer for pointing this out. We agree with the reviewer that the binding proteins can be repressors. In fact, we found that SNP rs17821573 (in Fig.2b) is located in a repression region. Compared with controls, the luciferase activity of the cloned fragments was significantly lower, indicating the repression effect of the cloned sequences.

Page 22 “... brain (FDR<0.05 in CMC, FDR<0.01 in LIBD, and P<0.001 in GTEx) ... ”

It is not justified to use different p-value thresholds in different data sets.

Response: We thank the reviewer for pointing this out and we are sorry for not mentioning why different P-value thresholds were used in different data sets. When we performing the eQTL annotation, we tried our best to follow the default parameters used in the original papers. In CMC data set, the default FDR threshold is 0.05. In LIBD data set, the default FDR is 0.01. In GTEx, we used P<0.001 as threshold. We could identify moderate targets genes at these thresholds. Lower threshold will increase the false positive rates. However, if a strict threshold was used, the results will be a bit conservative.

We added several sentences to address why different P-value thresholds in different data sets were used. Page 15, lines 15-17.

Page 23 “Compared with the random SNPs, the identified regulatory SNPs showed significant association with gene expression (Z=18.5, P=1.44E-76), ... ”

It might be useful to show a figure here.

Response: We appreciate the reviewer for this valuable suggestion. Following the reviewer’s suggestion, we added a new figure (Supplementary Fig. 6) in the supplementary data to show that the identified regulatory SNPs showed significant association with gene expression (Z=14.23, P=3.09E-46) compared with the random SNPs. Page 17, lines 5-9.

Figure 8 “... (P<1.0×10⁻⁷ for gene set in Table 1 and P<5.0×10⁻⁷ for gene set in Supplementary Table 8).”

The difference between Table 1 and Table S8 need to be clarified.

Response: We thank the reviewer for pointing this out. According to the reviewer’s suggestion, we have clarified the difference between Table 1 and Table S8 in the revised manuscript. Page 19, lines 11-21.

Page 29 “. CTCF is a transcriptional repressor encoded by CTCF gene, which has pivotal roles in transcription regulation. ”

Is the expression level of CTCF associated with any SNPs in cis? If so, those SNPs are expected to be associated with SCZ.

Response: We thank the reviewer for this valuable suggestion. Following the reviewer's suggestion, we performed eQTL analysis in the three brain eQTL data sets used in this study. We found that no SNPs were associated with CTCF in LIBD and CMC data sets. In GTEx data set, we found that 213 SNPs showed significant association with CTCF expression in the nucleus accumbens (basal ganglia) (Supplementary Table S11). Among the 213 SNPs associated with CTCF, rs75760574 showed the most significant association with CTCF expression. We thus examined the association between rs75760574 and schizophrenia in PGC2+CLOZUK data set (40,675 cases and 64,643 controls). We found that rs75760574 showed marginal significant association with schizophrenia ($P=0.002$). Of note, a recent study showed that *CTCF* is a novel risk gene for schizophrenia (Juraeva et al. 2014). Interestingly, Baruch et al. also found that SNPs in CTCF binding regions were associated with schizophrenia in the Jewish Ashkenazi population (Baruch et al. 2009). These results provide convergent evidence that support the potential role of *CTCF* gene in schizophrenia. However, more work is need to elucidate the role of CTCF in schizophrenia. **Page 25, lines 18-22, Page 26, line 1-7.**

*Page 31 "... using genotype data (379 individuals) from the 1000 Genomes Project (Phase I data, phase1_v3.20101123)."
Chinese, Europeans or all the 1000 Genomes samples?*

Response: We are sorry for the confusion. The genotype data used for linkage disequilibrium analysis were from Europeans. We have corrected this in the revised manuscript. **Page 29, lines 18-20.**

*Page 32 "As some of the credible causal SNPs from above three GWASs were overlapping, we removed the overlapping SNPs. A total of 23,400 non-overlapping ..."
A Venn diagram to show the overlaps?*

Response: We thank the reviewer for this valuable suggestion. According to the reviewer's suggestion, we have provided a Venn diagram in the revised manuscript to showing the overlapping SNPs (Supplementary Fig. 10). **Page 30, lines 4-6.**

*Page 36 "... selected 132 SNPs (the number of SNPs equals to the identified TF binding-disrupting SNPs) from PGC2 SNP list (9,444,230 SNPs in total) ..."
With minor allele frequencies of the SNPs matched to the 132 TF binding-disrupting SNPs?*

Response: We appreciate the reviewer for pointing this out. We did not match the minor allele frequencies of the SNPs in our original version. Following the reviewer's suggestion, we re-performed the analysis through matching the minor allele frequencies. Consistent with our previous result, we found that compared with the random SNPs, the identified regulatory SNPs showed significant association with gene expression ($Z=14.23$, $P=3.09E-46$)

(Supplementary Fig. 6), further supporting the potential regulatory effect of the identified TF binding-disrupting SNPs. **Page 35, lines 13-18.**

Reviewer #2 (Remarks to the Author):

The manuscript by Huo et al provides comprehensive, mostly bioinformatic analyses of a large number schizophrenia associated GWAS loci aiming to pinpoint the causal variant(s). The strength of the study is that for a few loci they have extended the bioinformatic analyses to wet lab validation experiments. The work has been performed quite comprehensively.

Response: We thank the reviewer for the positive comment on our work.

1) The main drawback with the current version of the manuscript is that it is not easy to extract what they actually found. Table 1 tries to summarize this, but is so comprehensive that it does not highlight the main findings. This should be clarified.

Response: We thank the reviewer for pointing this out. Following the reviewer's suggestion, we revised the manuscript accordingly to summarize the main findings. First, we added a new paragraph to highlight the main findings in the revised manuscript. Second, we also revised Fig. 1 accordingly to highlight the main findings. Third, we provided a new Figure (Fig. 9) to summarize the main results to complement the current Fig. 1 in the revised manuscript. **Page 20, lines 11-22, Page 21, lines 1-5, Revised Fig. 1 and Fig. 9.**

2) It is also hard to follow the number of associated SNPs, for example:

- a. On line 26 and 27 in the abstract (and line 347) they state that "We found 97 of the 132 TF binding-disrupting SNPs are associated with gene expression..."*
- b. On lines 134 and 135 they state that "In total, 66 prioritized top functional SNPs showed significant association with gene expression....". This is the only place in the manuscript where the number of 66 associated SNPs is mentioned*

Even after reading the manuscript several times it is hard to follow the numbers. I suggest that the authors prepare a figure/table that summarizes to the reader the main results to complement the current Fig 1 that demonstrates the flow of the analyses. Fig 1 clarifies nicely the flow of analyses, but it only partially clarifies how many SNPs at each point had a positive finding in expression analyses.

Response: We thank the reviewer for this valuable suggestion. According to the reviewer's suggestion, in the revised manuscript, we provided a new Figure (Fig. 9) that summarizes the main results to complement the current Fig. 1 that demonstrates the flow of the analyses. **Revised Fig. 1 and Fig. 9.**

3) The authors highlight the reporter gene assay studies, which are a good extension. However, understandably only 10 of the 66 SNPs (as stated on line 137, yet line 330 indicates that 9 SNPs were tested, please clarify) were tested in reporter assays. The abstract is misleading, it states on line 27 that “we validated the regulatory effect of the identified SNPs with reporter gene assays and..... “, this is an overstatement as only 10 (or 9, whichever is correct) was tested. The fact that only a small fraction of loci were validated should be stated more clearly throughout the manuscript.

Response: We thank the reviewer for pointing this out and we are sorry for the confusion. In fact, we tested 19 SNPs. We first annotated the potential causal SNPs with different annotation approaches (including CADD, Eigen, RegulomeDB, LINSIGHT and GWAVA) and prioritized 153 top (i.e., with the highest scores (CADD, Eigen, LINSIGHT and GWAVA) or smallest rating (RegulomeDB)) functional SNPs. Among the 153 prioritized top functional SNPs, 66 showed significant association with gene expression in human brain tissues. We thus selected 10 SNPs from these 66 prioritized functional SNPs for reporter gene assays. **Page 7, lines 8-17.**

We then utilized the functional genomics to identify the potential causal SNPs that disrupt binding of specific transcription factors (TFs). In total, we identified 132 SNPs that disrupted binding of 21 distinct TFs. We found that 97 out of the 132 TF binding-disrupting SNPs were associated with gene expression in human brain tissues. To test if the TF binding-disrupting SNPs have regulatory effects, we selected 9 SNPs from the 132 TF binding-disrupting SNPs and conducted reporter gene assays. Following the reviewer’s suggestion, we have revised the manuscript accordingly to make our statements more clearly. In addition, we also revised our manuscript accordingly to avoid overstatements. Finally, To avoid confusion, we highlight our main findings in Fig. 9 in the revised manuscript. **Page 2, lines 9-12, Page 14, lines 14-19, Fig. 9.**

4) The fourth main feature of the manuscript is that they only analyzed brain tissue. It is not surprising that they then find variants associated to neural tissue. This was a deliberate choice, which should be addressed more clearly in the text. In other words, it remains unclear how specific these findings are to neural tissues, compared to other tissues.

Response: We thank the reviewer for this valuable suggestion. Following the reviewer’s suggestion, we addressed why we used brain tissues more clearly in the revised manuscript.

“We focused on risk variants that have regulatory effect in human brain (or neuronal cells) as schizophrenia is thought to be a disorder that is mainly originated from dysfunction of brain function. First, previous studies showed that most schizophrenia risk genes (including genes involved in neurotransmission (e.g., DRD2, GRM3, GRIN2A, CACNA1C and CACNA1I)) identified by genetic studies play pivotal roles in brain (Schizophrenia Working Group of the Psychiatric Genomics Consortium. 2014; Pardinás et al. 2018). Second, accumulating evidence supports the neurodevelopmental hypothesis of schizophrenia (Fatemi and Folsom. 2009; Rapoport et al. 2012). Consistently, studies have showed that schizophrenia risk genes*

(including DISC1, RELN and GLT8D1) have important role in brain development through regulating proliferation and differentiation of neural stem cells (Mao et al. 2009; Senturk et al. 2011; Yang et al. 2018). More important, a recent study showed that schizophrenia associations were strongly enriched at enhancers active in brain tissues (Schizophrenia Working Group of the Psychiatric Genomics Consortium. 2014). These lines of evidence suggest that schizophrenia risk variants exert their effects mainly in brain tissues. Therefore, brain tissues (or neuronal cells) may represent the most relevant tissues for studying the effects of SCZ risk variants. We thus used Chip-Seq and eQTL data from brain tissues (or neuronal cells) to identify the potential causal SNPs and target genes regulated by the identified TF binding-disrupting SNPs.” Page 9, lines 5-18.*

In addition, to explore if the findings in this study are specific to neural tissues, we also compared the eQTL results derived from brain tissues and non-brain tissues (the results were listed in the table below). Briefly, we first explored the number of TF binding-disrupting SNPs that were associated with gene expression in brain tissues in GTEx. We found that 66 of the 132 TF binding-disrupting SNPs were associated with gene expression in brain tissues in GTEx. We then explored the number of SNPs associated with gene expression in other tissues. We selected 5 tissues, including artery, liver, ovary, prostate and spleen. The sample size of these tissues were comparable to the brain tissues. Among 132 TF binding-disrupting SNPs, only 17-33 SNPs showed significant association with gene expression in these five selected tissues. However, 66 SNPs showed significant association with gene expression in brain tissues. This result suggests that our findings may be specific to neural tissues. Nevertheless, more work is needed to address this question.

Table. The number of TF binding-disrupting SNPs that were associated with gene expression in different tissues

Tissue	Number of samples	Number of TF binding-disrupting SNPs	Number of eQTL SNPs
Artery-Coronary	152	132	24
Liver	153	132	17
Ovary	122	132	26
Prostate	132	132	16
Spleen	146	132	33
Brain	80~154	132	66

5) *It would be good to add a list of limitations of the study. Now it reads as slightly too comprehensive and does not give a reality check.*

Response: We thank the reviewer for this valuable suggestion. According to the reviewer’s suggestion, we have added a list of limitations of the study in the revised manuscript. **Page 27, lines 1-18.**

Reviewer #3 (Remarks to the Author):

In this manuscript, the authors aim to identify SNPs that have causal functions in relation to SCZ based on GWAS findings. The authors first prioritize SNPs based on either well-established annotations (CADD, Eigen, GWAVA, RegulomeDB and LINSIGHT) or TFBS using Chip-Seq data. They then test several prioritized SNPs by reporter gene assay to further support their findings. The authors claim that frequent disruption of CTCF is suggestive of an important role in schizophrenia.

The authors used a bioinformatics approach to integrate publicly available data sets with functional experiments to support their findings. However, the study seems largely based on arbitrary decisions without proper justification.

Response: We understand the reviewer's concern. Following the reviewer's suggestion, we have revised the manuscript accordingly to justify the aim of each analysis or experiment in the revised manuscript. Detailed response to each comment were found below.

In addition, this type of analysis, especially for SCZ, has been widely performed but the authors barely compare their work with previous findings. The reported findings from pathway/gene expression analyses are not novel.

Response: We thank the reviewer for pointing this out. According to the reviewer's suggestion, we have compared our results with previous finding in the revised manuscript. Page 24, line 22, Page 25, lines 1-6.

Below are my main comments followed by some minor comments.

1. Authors should clarify why they conducted two analyses with well-established annotations and Chip-seq, separately. This also makes the manuscript a little bit hard to follow.

Response: We thank the reviewer for this valuable suggestion. Following the reviewer's suggestion, we have clarified why we conducted two analyses with well-established annotations and Chip-Seq separately in the revised manuscript. Page 8, lines 7-22, Page 23, lines 8-22, Page 24, lines 1-8.

In addition, I believe some of the annotations such as CADD, RegulomeDB and GWAVA include Chip-seq data. I would expect to find overlap of prioritized SNPs between these two analyses but the authors did not discuss that. I would suggest to integrate both annotations as they can support each other. Otherwise the authors should clearly state why it is better to keep separate.

Response: We thank the reviewer for this valuable suggestion. According to the reviewer's suggestion, we compared the overlap of prioritized SNPs between the functional annotation and functional genomics in the revised manuscript. We also addressed why we performed functional annotation and functional genomics separately in the revised manuscript. Page 23, lines 8-22, Page 24, lines 1-8.

2. In the section of “Prioritization of functional SNPs and experimental verification”, the authors compared top functional SNPs per locus but it is not clear what are the “top functional SNPs”. It should be clarified, in the method section, what the scale of the scores is and the threshold for each annotation.

Response: We are sorry for the confusion. Following the reviewer’s suggestion, we have clarified the meaning of the “top functional SNPs” in the revised manuscript. **Page 6, lines 4-19.**

“Of note, two different strategies were used by the functional annotation methods to prioritize the potential functional SNP. For CADD, Eigen, GWAVA and LINSIGHT, the larger the score, the higher probability that the SNP is functional. Therefore, the SNP with the highest score was defined as the top functional SNP. For RegulomeDB, smaller rating suggests higher probability that the SNP is functional. Thus, the SNP with the smallest rating was defined as the top (i.e., the most likely) functional SNP. For CADD, Eigen, GWAVA and LINSIGHT, the SNP with the highest score at each locus was defined as top functional SNP. For RegulomeDB, the SNP with the smallest rating was defined as top functional SNP. For each locus, we compared the top functional SNPs identified by different annotation methods and found that 153 loci (PGC2(Schizophrenia Working Group of the Psychiatric Genomics Consortium. 2014) performed regular LD clumping (implemented in PLINK(Purcell et al. 2007)) to define the index SNPs (with following parameters: $r^2 < 0.1$, $P1 < 5 \times 10^{-8}$, window size < 500 kb) and the genomic region containing all SNPs that were in LD (i.e., $r2 > 0.6$) with each of the 128 index SNPs was defined as a locus) contained overlapping top functional SNPs prioritized by at least two different annotation methods (i.e., at least two methods annotated the same SNP as the most possible functional or causal SNP) (Supplementary Table 1-3), suggesting these SNPs were promising functional SNPs.”*

In addition, we also clarified the scale of the scores and the threshold for each annotation in the method section in the revised manuscript. Page 30, lines 13-22, Page 31, lines 1-16.

One concern is that, the score of “top SNPs” can vary highly per locus. For example, when the top score in locus A is 100, the top score in locus B can be 10 which is much less likely to be functional compared to locus A. The authors need to clarify how they dealt with this.

Response: We understand the reviewer’s concern and we agree with the reviewer that the score of “top SNPs” can vary highly per locus. However, as our goal is to identify the most possible functional SNP (i.e., have the highest scores or smallest ratings) at each risk locus, we did not compare the scores of the top functional SNPs from different loci. **Page 31, lines 13-16.**

3. Analyses related to fig. 2b: it is not clear how the authors selected the 10 SNPs.

Response: We thank the reviewer for pointing this out. We have provided the SNP selection criteria for reporter gene assays in the revised manuscript. **Page 7, lines 9-13.**

“The selection criteria of top functional SNPs for reporter gene assays were as follows: First, this SNP was prioritized as top functional SNP (has the highest score or the smallest rating) at least by two different annotation methods simultaneously. Second, this SNP was associated with gene expression in human brains (Supplementary Table 5).”

4. For 132 SNPs disrupting TFBSs, it is crucial to check their LD dependence. The authors should report how many of them are independent from 132. For example, they found 6 SNPs that had significant association with FAM109B but those SNPs can be in LD and those associations might thus not be independent.

Response: We thank the reviewer for this valuable suggestion. Following the reviewer’s suggestion, we performed LD analysis to investigate the linkage disequilibrium between the 132 TF binding-disrupting SNPs. We found that 40 TF binding-disrupting SNPs showed LD ($r^2 > 0.3$, using genotypes of Europeans) with other TF binding-disrupting SNPs (Supplementary Table 10). Of note, 8 TF binding-disrupting SNPs (rs10083370, rs2304206, rs3773744, rs4759413, rs60754073, rs76514049, rs78866909 and rs9616378) showed complete LD (i.e., $r^2 = 1$) with other TF binding-disrupting SNPs (Supplementary Table 10). These LD results indicate that some of the TF binding-disrupting were in linkage disequilibrium with each other. However, considering that all of these 132 SNPs disrupt binding of TFs, suggesting that these 132 TF binding-disrupting SNPs may have functional consequences. Thus, more work is needed to investigate if all of the highly linked TF binding-disrupting SNPs or only some of the highly linked TF binding-disrupting SNPs confer risk of schizophrenia through affecting gene expression. **Page 17, lines 13-22, Page 18, lines 1-6, Page 36, lines 3-8.**

5. Authors mentioned that POLR2A and CTCF have the largest number of SNPs but that may be because those TFs regulates more genes than other TFs. Authors should show if those findings are not random.

Response: We thank the reviewer for this valuable suggestion. According to the reviewer’s suggestion, we conducted additional analysis to address if the disruption of POLR2A and CTCF are random. The frequent disruption of CTCF and POLR2A binding (by the schizophrenia risk SNPs) may attribute to the possibility that CTCF and POLR2A regulate more genes than other TFs. To test if disruption of CTCF and POLR2A were random, we conducted additional analysis using the TRRUST (Transcriptional Regulatory Relationships Unraveled by Sentence-based Text mining from 11,237 pubmed articles) (Han et al. 2018), a manually curated database which contains about 800 human TFs and over 8000 TF-target genes regulatory relationships (see methods). We compared the number of target genes of CTCF and YY1 (POLR2A was not available) in the TRRUST database. We found that CTCF and YY1 have 36 and 105 target genes in the TRRUST database, respectively. If the frequent disruption of CTCF is due to the fact that CTCF regulates more genes than other TFs, the number of SNPs that disrupt YY1 binding will be equal or exceeds the number of SNPs that disrupt CTCF binding (as YY1 regulates more genes than CTCF). However, we

found that only 2 schizophrenia risk SNPs disrupt YY1 binding (Fig. 3a). These results suggest that the frequent disruption of CTCF may not be due to random effect. More work is needed to test if disruption of CTCF and POLR2A binding (by schizophrenia risk SNPs) was random or not. We added a new figure (Supplementary Fig. 9) in the revised manuscript. **Page 26, lines 8-22, Page 40, lines 4-15, Supplementary Fig. 9.**

6. Reporter gene assay was performed only for several SNPs from 132 SNPs. The authors should justify how they selected those SNPs and why they only tested these.

Response: We appreciate the reviewer for this valuable suggestion. According to the reviewer's suggestion, we clarified how we select the SNPs for reporter gene assays in the revised manuscript. **Page 14, lines 14-19.**

We also addressed why we only performed reporter genes assays for 19 identified functional SNPs in the revised manuscript. **Page 27, lines 8-12.**

“Third, considering that the reporter gene assays are relatively labor-consuming and time-costing (i.e., We needed to amplify the DNA fragments containing the test SNP first, we then cloned them into vectors. We used PCR-mediated point mutation technique to generate the DNA fragments containing the alternative allele of this test SNP. After validating the sequences of the inserted DNA fragments with Sanger sequencing, we transfected them into different cell lines for reporter gene assays.), we only tested 19 identified functional SNPs (10 SNPs from functional annotation and 9 SNPs from functional genomics). Thus, only a small proportion of the 132 TF binding-disrupting SNPs were verified with reporter gene assays.”

7. In the paper, the authors performed reporter gene assay to assess the regulatory function of prioritized SNPs but target genes are identified based on eQTLs annotations. eQTLs have been used previously to identify target genes from SCZ risk loci (e.g. Gusev, A. et al. Nat Genet. 2018), indeed findings from pathway enrichment and gene expression are already known (e.g. Pers, T. Human Mol. Genet. 2016, Won, H. et al. Nature. 2016). Authors should discuss what is novel and what is known already.

Response: We appreciate the reviewer for this valuable suggestion. Following the reviewer's suggestion, we compared our results with the target genes identified by Gusev et al. (Gusev et al. 2018) in the revised manuscript. **Page 24, lines 9-15.**

We also compared our pathway enrichment and gene expression analyses with the findings from Per et al. (Pers et al. 2016) and Won et al. (Won et al. 2016) in the revised manuscript. **Page 25, lines 1-13.**

Minor comments

1. The word “credible causal SNPs” is often used to refer to SNPs after in silico fine mapping. There is no official rule about this but I would advise to use slightly different wording for the

SNPs that are in LD with one of the GWAS hits, such as potential causal SNPs or candidate SNPs to avoid confusion.

Response: We thank the reviewer for this valuable suggestion. Following the reviewer's suggestion, we used potential causal SNPs (to replace the credible causal SNPs) throughout the revised manuscript to avoid confusion.

2. The section "Validation of the regulatory effect of the identified TF binding-disrupting SNPs" does not seem to contain any new information but is just a summary of the previous sections. I am not sure why this section is necessary.

Response: We are sorry for the confusion. In addition to the 7 SNPs (i.e., rs1801311, rs12912934, rs16937, rs7012106, rs2270363, rs7014953 and rs6992019) investigated in Fig. 4-7, regulatory effects of two additional SNPs (rs2535629 and rs2711116) were also tested with reporter gene assays (Supplementary Fig. 4 and 5). The results of rs2535629 and rs2711116 were not presented in previous sections. We thus listed the results of rs2535629 and rs2711116 in this section. In addition, as the reporter gene assays for different SNPs were sparsely distributed in different sections, we also wanted to draw a conclusion through combining the results of the reporter gene assays. Therefore, we listed these results in a separate section. We have revised this section accordingly in the revised manuscript. **Page 14, line 8.**

If the reviewer still feel it is not suitable for listing this section, we are glad to delete it or remove it into supplementary data.

3. On line 345, "However" does not seem to be a correct conjunction in that sentence.

Response: We thank the reviewer for pointing this out. We have corrected this in the revised manuscript. **Page 15, line 15.**

4. Authors mentioned "eQTL analysis" several times, but I assume publicly available summary statistics were used rather than performing actual eQTL analysis. To avoid confusion it should be rephrased to "eQTL annotation".

Response: We thank the reviewer for this valuable suggestion. Following the reviewer's suggestion, we have rephrased the "eQTL analysis" to "eQTL annotation" in the revised manuscript.

Again, we much appreciate the reviewers for their insightful and valuable comments and suggestions, which help us to improve the quality of the manuscript significantly. We hope that our revised work is satisfactory, and are happy to further improve it if needed.

References

- Baruch, K., Silberberg, G., Aviv, A., Shamir, E., Bening-Abu-Shach, U., Baruch, Y., et al. 2009. Association between golli-MBP and schizophrenia in the Jewish Ashkenazi population: are regulatory regions involved? *Int J Neuropsychopharmacol* 12 (7): 885-894.
- Fatemi, S. H. and Folsom, T. D. 2009. The neurodevelopmental hypothesis of schizophrenia, revisited. *Schizophr Bull* 35 (3): 528-548.
- Gilman, S. R., Chang, J., Xu, B., Bawa, T. S., Gogos, J. A., Karayiorgou, M., et al. 2012. Diverse types of genetic variation converge on functional gene networks involved in schizophrenia. *Nat Neurosci* 15 (12): 1723-1728.
- Gusev, A., Mancuso, N., Won, H., Kousi, M., Finucane, H. K., Reshef, Y., et al. 2018. Transcriptome-wide association study of schizophrenia and chromatin activity yields mechanistic disease insights. *Nat Genet* 50 (4): 538-548.
- Han, H., Cho, J. W., Lee, S., Yun, A., Kim, H., Bae, D., et al. 2018. TRRUST v2: an expanded reference database of human and mouse transcriptional regulatory interactions. *Nucleic Acids Res* 46 (D1): D380-D386.
- Juraeva, D., Haenisch, B., Zapatka, M., Frank, J., Witt, S. H., Muhleisen, T. W., et al. 2014. Integrated pathway-based approach identifies association between genomic regions at CTCF and CACNB2 and schizophrenia. *PLoS Genet* 10 (6): e1004345.
- Li, Z., Chen, J., Yu, H., He, L., Xu, Y., Zhang, D., et al. 2017. Genome-wide association analysis identifies 30 new susceptibility loci for schizophrenia. *Nat Genet* 49 (11): 1576-1583.
- Mao, Y., Ge, X., Frank, C. L., Madison, J. M., Koehler, A. N., Doud, M. K., et al. 2009. Disrupted in schizophrenia 1 regulates neuronal progenitor proliferation via modulation of GSK3beta/beta-catenin signaling. *Cell* 136 (6): 1017-1031.
- Pardinas, A. F., Holmans, P., Pocklington, A. J., Escott-Price, V., Ripke, S., Carrera, N., et al. 2018. Common schizophrenia alleles are enriched in mutation-intolerant genes and in regions under strong background selection. *Nat Genet* 50 (3): 381-389.
- Pers, T. H., Timshel, P., Ripke, S., Lent, S., Sullivan, P. F., O'Donovan, M. C., et al. 2016. Comprehensive analysis of schizophrenia-associated loci highlights ion channel pathways and biologically plausible candidate causal genes. *Hum Mol Genet* 25 (6): 1247-1254.
- Purcell, S., Neale, B., Todd-Brown, K., Thomas, L., Ferreira, M. A., Bender, D., et al. 2007. PLINK: a tool set for whole-genome association and population-based linkage analyses. *Am J Hum Genet* 81 (3): 559-575.
- Rapoport, J. L., Giedd, J. N. and Gogtay, N. 2012. Neurodevelopmental model of schizophrenia: update 2012. *Mol Psychiatry* 17 (12): 1228-1238.
- Schizophrenia Working Group of the Psychiatric Genomics Consortium*. 2014. Biological insights from 108 schizophrenia-associated genetic loci. *Nature* 511 (7510): 421-427.
- Senturk, A., Pfennig, S., Weiss, A., Burk, K. and Acker-Palmer, A. 2011. Ephrin Bs are essential components of the Reelin pathway to regulate neuronal migration. *Nature* 472 (7343): 356-360.
- Wang, K., Li, M. and Hakonarson, H. 2010. ANNOVAR: functional annotation of genetic variants from high-throughput sequencing data. *Nucleic Acids Res* 38 (16): e164.

- Won, H., de la Torre-Ubieta, L., Stein, J. L., Parikshak, N. N., Huang, J., Opland, C. K., et al. 2016. Chromosome conformation elucidates regulatory relationships in developing human brain. *Nature* 538 (7626): 523-527.
- Yang, C. P., Li, X., Wu, Y., Shen, Q., Zeng, Y., Xiong, Q., et al. 2018. Comprehensive integrative analyses identify GLT8D1 and CSNK2B as schizophrenia risk genes. *Nat Commun* 9 (1): 838.

Reviewer #1 (Remarks to the Author):

The authors have addressed all my previous concerns. I have no further comment.

Reviewer #2 (Remarks to the Author):

The authors have addressed all my critiques

Reviewer #3 (Remarks to the Author):

I am satisfied by the authors' responses except one point, for point 5, about a large number of SNPs disrupting POLR2A and CTFT binding sites. I might have not been clear enough about this in the last review, but what I meant was, if you have random 132 SNPs, how likely it is to find ~40 SNPs that are disrupting POLAR2 or CTFT binding sites. Rather than the number of target genes of POLR2A or CTFT, their binding sites on the genome should be the background in this case (you can either count the number of motifs appear on the genome or the total summed length of the motifs on the genome). It's possible that the proportion of binding sites of these TF is higher than other TFs, which make more likely to find SNPs within the binding sites.

Reviewers' comments:

Reviewer #1 (Remarks to the Author):

The authors have addressed all my previous concerns. I have no further comment.

Response: We thank the reviewer for his approval of our manuscript.

Reviewer #2 (Remarks to the Author):

The authors have addressed all my critiques.

Response: We thank the reviewer for his recommendation for publication of our manuscript.

Reviewer #3 (Remarks to the Author):

I am satisfied by the authors' responses except one point, for point 5, about a large number of SNPs disrupting POLR2A and CTFT binding sites. I might have not been clear enough about this in the last review, but what I meant was, if you have random 132 SNPs, how likely it is to find ~40 SNPs that are disrupting POLAR2 or CTFT binding sites. Rather than the number of target genes of POLR2A or CTFT, their binding sites on the genome should be the background in this case (you can either count the number of motifs appear on the genome or the total summed length of the motifs on the genome). It's possible that the proportion of biding sites of these TF is higher than other TFs, which make more likely to find SNPs within the binding sites.

Response: We are sorry that we misunderstood the 5th question of the reviewer. Following the reviewer's suggestion, we performed additional analyses. We found that binding of CTCF and POLR2A were frequently disrupted by schizophrenia risk SNPs. This may be due to the possibility that CTCF and POLR2A have more binding sites (on the genome) than other TFs. To test if disruption of CTCF and POLR2A were random, we conducted additional analyses. We first identified 8,447 SNPs that disrupted binding of 30 TFs (these 30 TFs were used in this study) through mapping a total of 968,903 SNPs (from the Illumina HumanOmni1-Quad) to the identified ChIP-Seq peaks (the identification of TF binding-disrupting SNPs was described in more detail above). We then randomly sampled 132 SNPs (the number of SNPs equals to the identified TF binding-disrupting SNPs in this study) from the 8,447 TF binding-disrupting SNPs and counted the number of the SNPs that disrupted the binding of each TF. We found that the number of POLR2A binding-disrupting SNPs observed in this study were significantly higher than random SNPs ($P=0.013$, 1,000 simulations, **Supplementary Fig. 9a), suggesting that the frequent disruption**

of POLR2A binding by schizophrenia risk SNPs may not due to random effect. However, the number of CTCF binding-disrupting SNPs (observed in this study) were not significantly different from random SNPs ($P>0.05$, 1,000 simulations, **Supplementary Fig. 9b**), implying that the frequent disruption of CTCF by schizophrenia risk SNPs may due to random effect. We also compared the number of motifs (appear on the genome) of the 30 TFs through counting the identified ChIP-Seq peaks of each TF. We found that RAD21, EP300 and POLR2A have the largest number of ChIP-Seq peaks on the genome (**Supplementary Fig. 9c**). In addition, the number of ChIP-Seq peaks of CTCF were also larger than many other TFs (including GATA3, REST, TAF1 and YY1). These results suggest that schizophrenia risk SNPs tend to disrupt POLR2A binding (but not CTCF). However, more work is need to further verify this.

We have included a new figure (**Supplementary Fig. 9**, please see below) to reflect these results. **Page 26, lines 8-22, Page 27, lines 1-6, Page 40, lines 11-22, Page 41, lines 1-5.**

Figure S9. The number of random SNPs that disrupt POLR2A and CTCF binding. (a) The distribution of random SNPs that disrupted POLR2A binding. The number of POLR2A

binding-disrupting SNPs observed in this study were significantly higher than random SNPs ($P=0.013$, 1,000 simulations). **(b)** The distribution of random SNPs that disrupted CTCF binding. The number of CTCF binding-disrupting SNPs observed in this study were not significantly different from random SNPs ($P>0.05$, 1,000 simulations). **(c)** The number of motifs (ChIP-Seq peaks) of each TF included in this study. The detail information about the ChIP-Seq peaks from different neuronal cells or brain tissues can be found in Supplementary Table 6.

Again, we much appreciate the reviewers for their insightful and valuable comments and suggestions, which help us to improve the quality of the manuscript significantly. We hope that our revised work is satisfactory, and are happy to further improve it if needed.

Reviewer #3 (Remarks to the Author):

The authors have addressed all my concerns.

Reviewers' comments:

Reviewer #3 (Remarks to the Author):

The authors have addressed all my concerns.

Response: We thank the reviewer for his/her recommendation for publication of our manuscript.